# Molecular Characterization of Polar Organic Aerosol Constituents in Off-Road Engine Emissions Using Fourier Transform Ion Cyclotron Resonance Mass Spectrometry (FT-ICR MS): Implications for Source Apportionment

Min Cui[1,2]; Cheng Li[3]; Yingjun Chen[4,1*]; Fan Zhang[1]; Jun Li[2*]; Bin Jiang[2]; Yangzhi Mo[2]; Jia Li[5]; Caiqing Yan[6]; Mei Zheng[6]; Zhiyong Xie[7]; Gan Zhang[2]; Junyu Zheng[3*]

[1]College of Environmental Science and Engineering, Tongji University, Shanghai 200092, P.R. China

[2]State Key Laboratory of Organic Geochemistry, Guangzhou Institute of Geochemistry,

Chinese Academy of Sciences, Guangzhou, 510640, P.R. China

[3]Jinan University Institute for Environmental and Climate Research, Guangzhou, 511443, P.R. China

[4]Shanghai Key Laboratory of Atmospheric Particle Pollution and Prevention (LAP3), Department of Environmental Science and Engineering, Fudan University, Shanghai

200433, P.R. China

[5]School of Environmental Science and Engineering, Yangzhou University, Yangzhou 225127, P.R. China

[6]SKL-ESPC and BIC-EAST, College of Environmental Sciences and Engineering, Peking University, Beijing 100871, P.R. China

[7]Helmholtz-Zentrum Geesthacht, Centre for Materials and Coastal Research, Institute of Coastal Research, Geesthacht, 21502, Germany

*Corresponding authors: Yingjun Chen (yjchenfd@fudan.edu.cn); Jun Li

(junli@gig.ac.cn); Junyu Zheng (zhengjunyu_work@hotmail.com)

**Abstract:** The molecular compositions of polar organic compounds (POCs) in particles emitted from various vessels and excavators were characterized using Fourier Transform Ion Cyclotron Resonance Mass Spectrometry (FT-ICR MS) and possible molecular structures of POCs were proposed. POCs were extracted with purified water and sorted by elemental composition into three groups: CHO, CHON and S-containing compounds (CHONS and CHOS). The results show that: (i) CHO (accounting for 49% of total POCs in terms of peak response) was the most abundant group for all tested off-road engines, followed by CHON (33%) and CHOS (35%) for diesel and HFO (heavy fuel oil) -fueled off-road engines, respectively. (ii) The abundance and structure of the CHON group in water extracts were different in terms of engine type and load. The relative peak response of CHON was the highest for excavator emissions in working mode, compared to the idling and moving modes. Furthermore, dinitrophenol and methyl dinitrophenol were potentially the most abundant emission species for high-rated speed excavators, while nitronaphthol and methyl nitronaphthol were more important for low-rated speed vessels. (iii) The composition and structure of the S-containing compounds were directly influenced by fuel oil characteristics (sulfur content and aromatic ring), with more condensed aromatic rings in the S-containing compounds proposed in HFO-fueled vessel emissions. More abundant aliphatic chains were inferred in diesel equipment emissions. Overall, higher fractions of condensed hydrocarbons and aromatic rings in POCs emitted from vessels using HFO cause strong optical absorption capacity. Different structures in POCs could provide a direction for

qualitative and quantitative analysis of organic compounds as tracers to distinguish these emissions from diesel or HFO-fueled off-road engines.

## 1. Introduction

A rapid increase in the number of off-road engines (e.g. vessels and excavators) has resulted in large quantities of pollutants emission, which have severe impacts on air quality, human health and climate change (Righi et al., 2011; Li et al., 2016; Liu et al., 2016; Wang et al., 2018; Zhang et al., 2018). In China alone, the dead weight capacity of vessels increased from 51 million tons in 2000 to 266 million tons in 2016 (NBS 2017). It was reported that emissions from fishing boats accounted for 18.3% of total fine particulate matter ($PM_{2.5}$) emitted by on-road vehicles (Deng et al., 2017). Almost 14, 500-37, 500 premature deaths per year were caused by emissions from ocean-going vessels in East Asia (Liu et al., 2016). As reported by the US EPA, nearly 34% of elemental carbon (EC) emissions originated from off-road diesel vehicle emissions in the United States (USEPA 2015). Furthermore, construction equipment, a typical off-road diesel vehicle, emitted abundant volatile organic matter (VOC). VOC was considered one of the most important precursor of HUmic-LIke Substances (HULIS) like organosulfates (Zhang et al., 2010; Tao et al., 2014).

Emission standards for off-road engines are not fully implemented in China, especially for vessel emissions. Currently, stage 3 emission standard has been implemented for off-road diesel engines since 2016, while stage 1 emission standard for emission from vessels will be implemented in 2020 (SEPA et al., 2015a; SEPA et al., 2016). Furthermore, the oil quality for off-road mobile sources cannot be guaranteed.

According to the standard of GB/T17411-2012, the sulfur content in oil used for vessels could reach 1-3.5%, which was 200-700 times higher than those for China IV diesel (SEPA et al., 2015b). There is a continued need to apportion the contributions of off-road engines to atmospheric $PM_{2.5}$. However, to the best of our knowledge, there is no unique tracer available to identify off-road engines combustion (Zhang et al., 2014; Liu et al., 2017a). According to published studies, vanadium (V) was usually used to identify the contribution of $PM_{2.5}$ from ship emission. However, it should be noted that V emissions can also be attributed to industrial activities. Therefore, overestimated/underestimated results of contributions from ship emissions to $PM_{2.5}$ in the atmosphere are obtained while using V as the only tracer (Zhang et al., 2014). Furthermore, some isomer ratios of polycyclic aromatic hydrocarbons were recommended as indicators of off-road engine emissions, but the huge variations of these ratios in the atmosphere might affect the end results (Zhang et al., 2005; Cui et al., 2017). Thus, there is an urgent need to explore unique organic tracers.

Organic matter (OM) is one of the most important components in $PM_{2.5}$ emitted by off-road vehicles, with the highest fraction accounting for up to approximately 70% in ship emissions (Cui et al., 2017). However, substantial portions of OM emitted from off-road engine combustion, especially for POCs, were difficult to isolate and identify by traditional analytical instruments, and still remain unknown. For example, HULIS was reported to arise primarily from biomass burning and fossil fuel (coal and diesel) combustion. There is substantial literature that deals with the optical characterization and molecular composition of HULIS emitted from biomass burning, an important

source of BrC worldwide. However, the contribution of diesel combustion to BrC was also controversial (Zheng et al., 2013). A reason for these discrepant degrees of recognition between diverse sources was the similarity in chemical structure (nitrogen-containing bases and nitroaromatics) found between HULIS and the compounds emitted from biomass burning (Ren et al., 2018; Violaki and Mihalopoulos 2010). It may also be due to a lack of knowledge on characterization of POCs emitted from diesel vehicles, especially for off-road diesel vehicles.

FT-ICR MS is an advanced technique with a high mass resolution of 0.00001 and is commonly used to determine the organic matter composition at a molecular level in crude oil (Jiang et al., 2019). FT-ICR MS is usually coupled with soft ionization techniques, such as electrospray (EST) and atmospheric pressure chemical ionization (APCI). They are used to analyze polar species and non-polar organic matters, respectively (Smith et al., 2009; Smit et al., 2015). It should be noted that FT-ICR MS, without chromatographic separation, can only detect molecular formulas and molecular identification based on elemental composition alone. This is challenging because most complex molecules have several stable isomeric forms (Laskin et al., 2009). However, some traditional mass spectrometry methods are equipped with quadrupole, ion trap, or time of flight, which have limited resolving power compared with the FT-ICR MS. Recently, some studies have successfully characterized the elemental components of polar organic compounds present in the atmosphere or emitted by different sources using FT-ICR MS (Wozniak et al., 2008; Laskin et al., 2009; Smith et al. 2009; Yassine et al. 2014). A few of them have been undertaken in China, especially for source

emission (Lin et al., 2012; Jiang et al., 2016; Mo et al., 2018; Song et al., 2018). Song et al. (2018) reported that the most abundant group of HULIS emitted from biomass burning and coal combustion was CHO, followed by CHON for biomass burning and CHOS for coal combustion. In contrast, Wang et al. (2018) observed that CHON was the dominant compound emitted from straw residue burning. In addition, the possible chemical structure of HULIS could be determined by FT-ICR MS. Tao et al. (2014) compared the molecular compositions of organosulfates in aerosols sampled in Shanghai and Los Angeles. They found that the organosulfates in Shanghai had a low degree of oxidation and unsaturation, indicating the presence of long aliphatic carbon chains. Smith et al. (2009) reported that organic aerosol emitted from biomass burning showed a clear trend of increasing saturation with increasing molecular weight and exhibited a significant $CH_2$-based homologous series.

This study aimed to speculate chemical characterization of polar organic aerosol constituents at the molecular level emitted from typical off-road engines by FT-ICR MS. To this end, studies were conducted: 1) to identify the molecular composition of POCs from excavators under three different operation modes; 2) to determine the molecular composition and possible structure of POCs from vessels using HFO and diesel, respectively; 3) to explore the key factors affecting the composition and structure of POCs from HFO and diesel fueled off-road engines; 4) to pave the way for the discovery of potential tracers for off-road engine emissions.

## 2. Materials and Methods

### 2.1 Sample collection

Four ships using HFO and diesel, and four excavators covering different emission standards and powers were chosen as representative off-road engines in China. Detailed information about the four ships and four excavators is presented in **Table 1**. Before conducting field sampling, the original fuel was obtained directly from the fuel tank

and sent to the testing company for quality evaluation (**Table 2**). For excavator emission sampling, three operation modes (idling, moving, and working) were selected, and sampling time was approximately half an hour for each mode. The flowrate of the $PM_{2.5}$ sampler was 10 L/min. Commercial equipment (MFD25, produced by Shanghai Besser Environmental Protection Technology Co., Ltd.) was used for $PM_{2.5}$ sampling. The

description of the particulate matter dilution and sampling system are presented in Xia's published study (Xia, 2017). In short, the exhaust plume was pumped into a retention chamber and $PM_{2.5}$ was intercepted by the four $PM_{2.5}$ samplers. One Teflon and three quartz filters ($\Phi$=47 mm) were finally acquired and one of the quartz filters was used to determine the chemical composition of $PM_{2.5}$ for each excavator under the three

operation modes. For vessel emissions, an on-board measurement system was used, as previously described (Cui et al., 2017; Deng et al., 2017). The on-board measurement system contains one dilution tunnel connected to two particulate samplers. Finally, particulate matter was collected on two quartz filters ($\Phi$=90 mm) for each vessel.

**Table 1 Technical parameters of test off-road engines**

| Vehicle ID | Engine power (kW) | Type | Length × width (m) | Material | Age (years) | Rated speed (rpm) | Fuel type |
|---|---|---|---|---|---|---|---|
| YK | 4440 | vessel | 116×18 | Metal | 11 | 173 | HFO |
| YF | 5820 | Cargo vessel | 139×20.8 | Metal | 16 | 141 | HFO |
| GB1 | 91 | Gillnet | 20×4.3 | Wooden | 10 | 1500 | Diesel |
| TB4 | 235 | Trawler | 24×5.2 | Wooden | 7 | 1310 | Diesel |
| CAT320 | 106 | Excavator | 9.5*3.2 | Metal | >11 | 1650 | Diesel |
| CAT330B | 165.5 | Excavator | 11.1*3.3 | Metal | >11 | 1800 | Diesel |
| CAT307 | 85 | Excavator | 6.1×2.3 | Metal | 9 | 2200 | Diesel |
| PC60 | 40 | Excavator | 6.1*2.2 | Metal | 9 | 2100 | Diesel |

**Table 2 Results of the fuel quality analysis**

| Engine ID | Carbon (C) % | Hydrogen (H) % | Oxygen (O) % | Nitrogen (N) % | Sulfur (S) % | Vanadium (V) mg/kg | Water Content MJ/kg | Kinematical viscosity (40℃) mm$^2$/s |
|---|---|---|---|---|---|---|---|---|
| YK | 84.12 | 10.38 | 4.26 | 0.79 | 0.448 | 5 | 0.21 | 123.2 |
| YF | 80.54 | 10.05 | 8.23 | 0.78 | 2.46 | 19 | 8.98 | 410.2 |
| GB1 | 85.96 | 12.76 | <0.3 | 0.49 | 0.022 | / | / | 4.517 |
| TB4 | 86.21 | 12.47 | 0.45 | 0.49 | 0.323 | / | / | 4.976 |
| CAT320 | 86.38 | 11.5 | 2.00 | 0.05 | 0.019 | <1 | Trace | 5.592 |
| CAT330B | 86.38 | 11.5 | 2.00 | 0.05 | 0.019 | <1 | Trace | 5.592 |
| CAT307 | 86.32 | 11.2 | 1.99 | 0.05 | 0.138 | <1 | Trace | 5.420 |
| PC60 | 85.88 | 12.1 | 1.85 | 0.04 | 0.034 | <1 | Trace | 4.782 |

**2.2 Chemical analysis**

Due to the limitations of organic matter load in filters and cost-prohibitive analysis, the filters sampled from off-road engines with the same operation modes or fuel quality were combined together to characterize the comprehensive molecular compositions of POCs for off-road engines under different operation modes and fuel quality. As shown in **Table S1**, five samples (1, 2, 3, 4 and 5) were selected to conduct FT-ICR MS analysis, which represented vessels using heavy fuel oil, vessels using diesel,

excavators under idling, moving, and working modes, respectively. Sample 1 was combined with 25% of the filter area from the two HFO-fueled vessels, namely YK and YF; sample 2 was combined with 25% of filter area from two diesel-fueled vessels, namely GB1 and TB4; samples 3, 4, and 5 were combined with 50% of the filter areas from four excavators under idling, moving, and working modes, respectively, namely CAT320, CAT330B, CAT307 and PC60. The portions of filters (**Table S1**) were cut and combined for 40 min, subjected to ultrasonic extraction with 40 mL ultrapure water, and then filtered using a 0.22 μm PTFE membrane (Jinteng, China). The extraction solvent was then divided into three portions. Two portions were used for measuring the concentrations of organic carbon and optical absorbance, as described in the Supporting Information part A and B. The third portion was processed to assess the chemical composition of POCs by FT-ICR MS. The remaining extraction filters were frozen and then dried to remove ultrapure water. The filters were then subjected to 40 min of ultrasonic extraction with 36 mL dichloromethane and 4 mL methanol. The extracted solvent was divided into three portions, and these were processed in the same way as previously described for the ultrapure water extract.

Both samples extracted with water or organic solvents were processed by a solid phase extraction (SPE) method to remove ions, which disturbed the results of FT-ICR MS. A majority of inorganic ions (e.g. ammonium, sulfate, and nitrate) and low-molecular-weight organic compounds such as isoprene-derived organosulfates and sugars could be removed during SPE treatment (Gao et al. 2006, Lin et al. 2012, Surratt et al. 2007), which were not discussed in this research. The details of the solid phase

extraction method were presented by Mo et al. (2018). Briefly, the pH value of water extracts was adjusted to 2.0 using HCl, and then passed through an SPE cartridge (Oasis HLB, 30 μm, 60 mg/cartridge, Waters, USA). The adsorbed POCs were eluted with 6 mL 2% (v/v) ammonia/methanol and dried under a gentle stream of $N_2$. Finally, the POCs were re-dissolved using 10 mL ultrapure water.

## 2.3 FT- ICR MS analysis

The molecular characterization of POCs was undertaken using negative-ion ESI FT-ICR MS (Bruker Daltonics GmbH, Bremen, Germany) with a 9.4-T refrigerated actively shielded superconducting magnet. Extracted solutions were injected at flow rate of 180 $\mu L \cdot h^{-1}$ through an Apollo II electrospray source. Emitter voltage, capillary column introduction voltage, and capillary column end voltage for negative-ion formation were 3.0 kV, 3.5 kV and -320 V, respectively. The scan range was m/z 100−900 with a resolution >450 000 at m/z = 319 with <0.4 ppm absolute mass error. During analysis, nitrogen-containing compounds were used as an internal calibration. Finally, the spectrum peaks with ratio of signal/noise higher than 10 were exported.

C$c$H$h$O$o$N$n$S$s$ was used as a general formula, since some criteria should be conformed to assign the possible formula (Wang et al., 2017). Briefly, all of the mathematically possible formulas for each ion were calculated with a mass tolerance of ±2 ppm. The H-to-C, N-to-C, O-to-C, and S-to-C ratios were limited to 0.3−3.0, 0−0.5, 0−3.0, and 0−2.0, respectively, in the ESI⁻ mode. Peak magnitude is not indicative of a compound's concentration in a sample due to inherent biases of SPE extractions and electrospray ionization efficiencies (Wozniak et al., 2008). Therefore, the relative

responses of detected peaks are discussed here. Due to the common occurrence of contamination during ESI analysis (Smit et al., 2015), the relative response of all peaks was calibrated by subtracting the response of peaks detected for blank filters. The double bond equivalents (DBE) and aromaticity equivalents (Xn) were calculated as follows:

$$DBE=1+1/2\ (2c-h+n) \tag{1}$$

$$Xn=(3*(DBE-o-s)-2)/(DBE-o-s) \tag{2}$$

Where c, h, o, n, and s were the number of C, H, O, N, and S atoms in the corresponding formulas. It should be noted that the formula with DBE<0 or Xn<0 has been excluded (Wang et al., 2017).

## 3. Results and Discussion

### 3.1 General characteristics of POCs for off-road engines

In general, the range of detected peaks for excavators and vessels had molecular weights between 150-900 Da, but most of the intensive peaks occurred in the molecular weight range of 200-400 Da. The mass spectra for excavators in different operational modes and vessels using different oils were varied from each other. The number of peaks for POCs were 4734, 3097, 4731, 4554 and 2818, in excavator emissions under the idling, working, and moving modes, and vessel emissions using HFO and diesel, respectively. The average molecular weight of excavator emissions under the working mode and vessels using HFO were the lowest (322.6 ± 69.9 Da and 331.3 ± 72.9 Da respectively).

For excavators, CHO was the most abundant group of POCs in all three operation

modes, accounting for 41%, 46% and 48% of all the formulas in terms of peaks response for the idling, working, and moving modes, respectively. S-containing compounds (CHOS and CHONS) were most abundant in the idling mode, while the relative peak response of the CHON group was highest under the working mode (**Fig. 1**). For vessels, CHO was the most abundant species group of POCs for both the vessel using diesel and the ones using HFO, accounting for 50-60% of total peak intensity. However, CHOS accounted for almost 30% of total ion intensity for vessels using HFO, higher than other off-road diesel engines. Furthermore, the chemical properties of POCs for vessels using HFO showed a larger degree of oxidation and unsaturation than other samples (**Table S2**). These differences in the composition of POCs might be attributable to variations in engine load, fuel supply, and air supply in different operation modes, which are discussed later.

As discussed in supporting information (SI Part C), the chemical properties of extractions derived from water or DCM/MeOH were significantly different (**Fig. S1 and Fig. S2**). And through comparing the optical properties between water and DCM/MeOH extractions, it was found that the average mass absorption efficiency of water extracts was significantly higher than those for 90% DCM+10% MeOH extracts (**Fig. S3**). Thus, it was necessary to extract $PM_{2.5}$ by water to explore the emission characteristics of POCs from off-road engines.

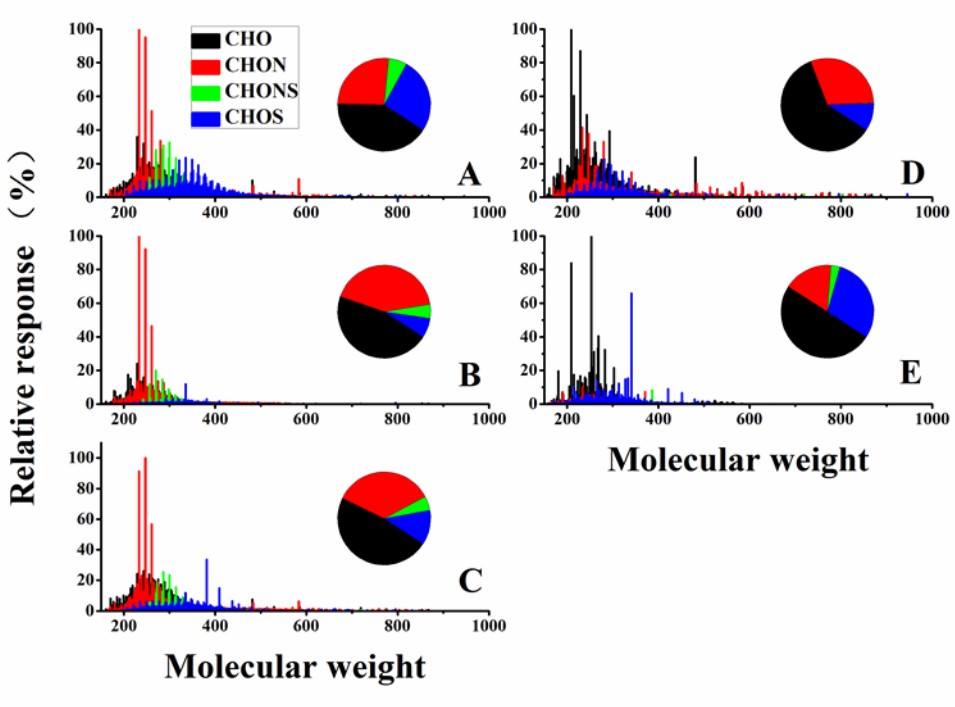

**Figure 1** Mass spectra of POCs in water extractions for off-road diesel engines emissions. A, B, C, D and E were the mass spectra for excavators under idling, working, moving modes, and vessels using diesel and HFO, respectively

## 3.2 CHO compounds in POCs from off-road engines emission

The number of peaks for CHO compounds were 1746, 1287, 1797, 1561, and 1318 for excavators in the idling, working and moving modes, and vessels using HFO and diesel, respectively. Considering the number of detected peaks for CHO compounds, the compositions of CHO group emitted from off-road engines were more complicated than those from ambient samples while being relatively comparable to those from other sources of emission (e.g. biomass: 1514-2296; coal combustion: 918) (Lin et al., 2012; Jiang et al., 2016; Song et al., 2018).The average molecular weight of detected ions for CHO compounds for excavators in idling, working and moving and vessels using HFO and diesel were 338 ± 96.7, 316 ± 84.6, 336 ± 96.6, 331 ± 72.9 and 357 ± 123 Da,

respectively, which were significantly higher than those emitted by coal and biomass burning (m/z=227-337 Da) (Song et al., 2018). Excavators under working mode had higher engine loads and combustion temperatures than those in other operation modes. Thus, the lowest number of CHO group ions and smallest average molecular weight were found during the working mode compared to the idling and moving modes. This indicated that long chain aliphatic hydrocarbons were liable to crack under elevate temperatures and low air/fuel ratio conditions. This trend was consistent with the results of a previous study (Wang et al., 2018) which found that low temperatures and oxygen-rich combustion would promote a chain propagation reaction. Although the lipid contents of fossil fuel might be an important precursor of CHO compounds from off-road engines, different fractions of heteroatom and isomers could lead to significantly different structures for the CHO group (Hellier et al., 2017). The highest intensity peaks for CHO compounds for off-road diesel engines were $C_9H_5O_6$ and $C_{13}H_9O_4$, while for vessels using HFO, the highest peaks were $C_{10}H_5O_8$, $C_9H_5O_6$ and $C_{10}H_5O_9$.

The Van Krevelen (VK) diagram (H/C versus O/C) was generally used to identify the structural properties of organic matter in FT-ICR MS research as only the molecular formula was given (Wozniak et al., 2008; Lin et al., 2012). Upon comparing the ratios of H/C and O/C for CHO compounds for different off-road engines under three operational modes and using different fuel oils, we concluded that the CHO group for vessels using HFO had the highest degree of oxidation and unsaturation. Furthermore, the CHO group in the working mode had a higher degree of oxidation and unsaturation than in idling and moving modes. As shown in **Fig. 2**, region 1 possibly represented

monocarboxylic acid, which was more abundant in both idling and moving modes than in the working mode (Wozniak et al., 2008; Lin et al., 2012). Region 2 represented compounds with low ratios of H/C and O/C and DBE>10 which were commonly speculated to consider as condensed hydrocarbons (Wozniak et al., 2008; Lin et al., 2012). Most compounds detected in the CHO group for vessels using HFO were molecular species in region 2 with a high number of O atoms and a low ratio of H/C. This was consistent with the original structure of combustion HFO, which was defined as bottom residue oil, containing fewer aliphatic hydrocarbons than those for diesel (Wikipedia, 2018). Furthermore, low engine speeds for vessels using HFO caused low temperature combustion which was prone to addition of $O_2$ to alkyl radicals and the subsequent formation of 6-member ring isomers (Sarathy et al., 2011; Ranzi et al., 2015). The ratios of DBE/C can be used as an indicator for condensed aromatic ring structures (Hockaday et al., 2006; Lin et al., 2012; Yassine et al., 2014). When DBE/C was higher than 0.7, compounds were commonly identified as soot-materials or oxidized polycyclic aromatic hydrocarbons (PAHs), an important class of light-absorption organic materials. The relative response of compounds with DBE/C>0.7 accounted for 3.2%, 6.5%, 3.1%, 26% and 8.3% of total ions for excavators under the idling, working and moving modes and vessels using HFO and diesel, respectively.

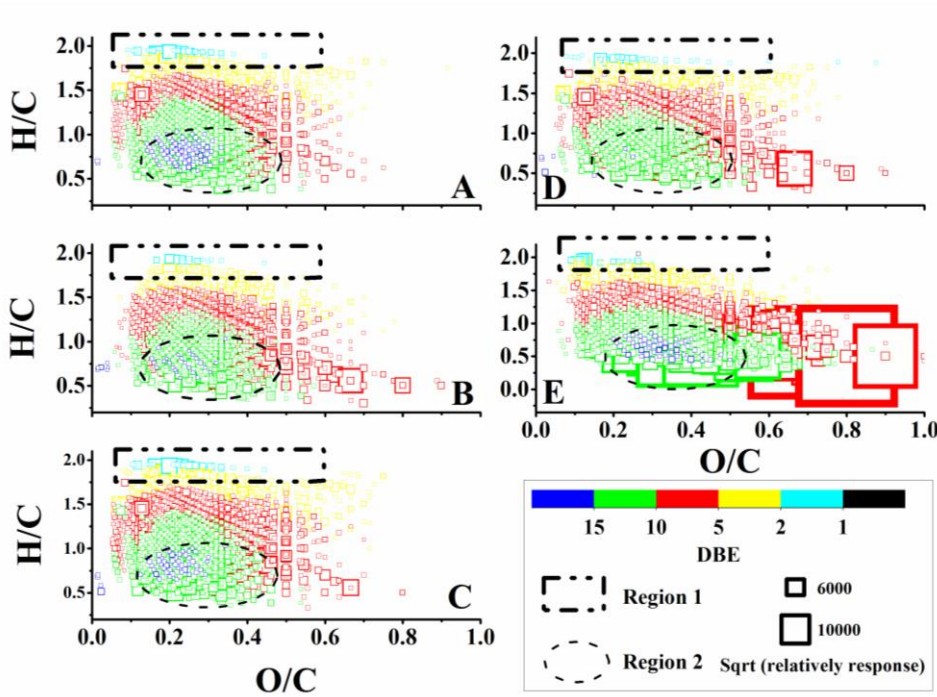

**Figure 2** The Van Krevelen (VK) diagrams of CHO compounds for off-road engines. (A, B, C, D and E were the mass spectra for excavators under idling, working, and moving modes and vessels using diesel and HFO, respectively)

## 3.3 CHON compounds in POCs from off-road engines emission

The peaks intensity percentage for the CHON group to total ions was the second largest in POCs emitted from off-road diesel engines, except for vessels using HFO (**Fig. 1**). The fraction of nitrogen oxide declined with increasing length of the straight-chain alkyl (Hellier et al., 2017), which was consistent with the relative response of the CHON group for diesel and HFO fueled engine emissions. It was always considered that CHON mainly originated from biomass burning emission (18%-41%), while the percentage of peaks response for the CHON group to the total assigned ions measured from off-road diesel engines was comparable or slightly smaller than those emitted from biomass burning (Laskin et al., 2009; Wang et al., 2017; Song et al., 2018). As shown by the average ratios of O/C and H/C for CHON group for vessels and

excavators, the degree of oxidation and unsaturation of the CHON group for vessels were higher than those for excavators (**Table S2**).

$C_{10}H_5O_5N_2$, $C_{11}H_7O_5N_2$, and $C_{12}H_9O_5N_2$ were the most abundant peaks in terms of relative responses for the CHON group detected in diesel fueled excavators and vessel emissions, while $C_{10}H_4NO_6$, $C_9H_4NO_4$, and $C_{10}H_4NO_7$ were highest for vessels using HFO. Diesel-fueled off-road equipment and vessels using HFO were high-rated speed and low-rated speed engines, respectively (**Table 1**). This might be the main reason for the formation of 2 nitrogen atoms in the CHON group for high-rated speed equipment, which results in higher combustion temperatures than those for low-rated speed engines, thereby promoting nitrogen atom attachment. The higher oxygen content in vessel emissions using HFO might be attributable to the higher oxygen content and kinematical viscosity of the HFO (**Table 2**). The large value of kinematical viscosity in HFO was a result of the presence of a certain number of aromatic hydrocarbons, fatty acids etc. which were difficult to combust and resulted in incomplete combustion (Örs et al., 2018).

For further discussion of possible chemical structures, the CHON group was divided into 23 subgroups, including OxN1 ($1 \leq x \leq 10$) and OyN2 ($2 \leq y \leq 14$) (**Fig. S4**). The distribution patterns of CHON subgroups for excavators under three operational modes were similar, with the highest relative response of $N_2O_5$, while $NO_4$-$NO_5$ were the highest group for vessel emissions in terms of relative response. The ratio of O/N higher than 3 is always indicative of the presence of nitro compounds (-$NO_2$) or some organic nitrates (with $NO_3$). Yassine et al. (2014) reported that DBE/C was only

valuable to the aromaticity properties of compounds with pure hydrocarbons.

The aromaticity equivalent (Xn) has been proposed to evaluate the aromaticity of organic material with heteroatoms (e.g. N, S). When the value of Xn exceeds 2.5, aromatic structures are most likely to be present within the compounds, while a value of Xn higher than 2.7, indicates the presence of condensed aromatic compounds (e.g. benzene core structure with Xn =2.5; pyrene core structure with Xn=2.83; ovalene core structure with Xn=2.92). As mentioned, the formulas in CHON group with the most abundant relative responses for diesel-fueled excavators and vessels were $C_{10}H_5N_2O_5$, $C_{11}H_7N_2O_5$, and $C_{12}H_9N_2O_5$, which compose the largest green ball in **Fig. 3** with Xn=2.5, it most likely indicated the presence of a benzene core structure in the compounds. Thus, it was could be dinitrophenol, and methyl dinitrophenol compounds. Likewise, $C_{10}H_4NO_6$, $C_9H_4NO_4$ and $C_{10}H_4NO_7$ comprise the largest yellow ball in **Fig. 3** for HFO-fueled vessels, most of which have Xn>2.7 indicating the presence of condensed aromatic compounds. Nitronaphthol and methyl nitronaphthol were potentially the most significant compounds arising from HFO-fueled vessel emissions, which have previously been reported in vehicle emissions (Yassine et al., 2014; Tong et al., 2016). Furthermore, almost 55% of the CHON group had an O/N ratio higher than 5, and half of the CHON group had aromatic rings higher than 3 for HFO-fueled. This was significantly higher than those for excavators and vessels using diesel. The reactivity and life span of these compounds should be considered to determine whether these chemicals could be used as tracers for off-road engines combustion. The same chemical structure discovered in the atmosphere indicated that secondary organic

aerosol (SOA) could provide valid evidence that CHON with nitrophenol or nitronaphthol could exist in the atmosphere long enough to be detected (Zhang et al., 2010).

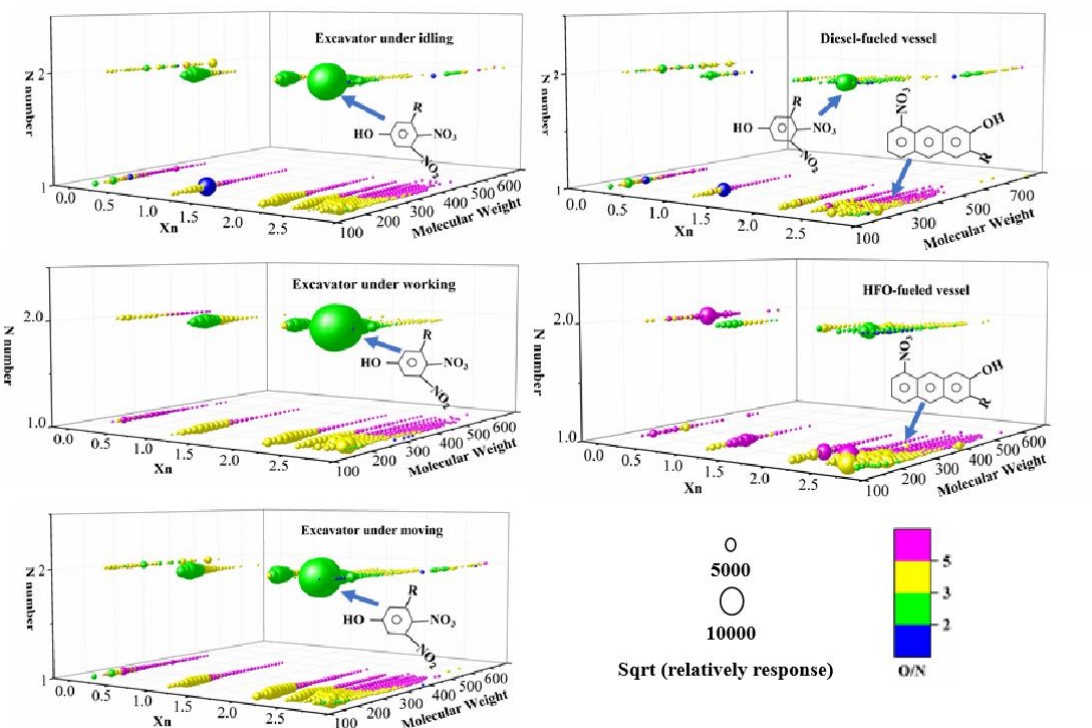

**Figure 3** Molecular composition and possible structure of CHON for excavators under three modes and vessels using HFO and diesel

## 3.4 S-containing compounds in POCs from off-road engines emission

The last group of POCs was S-containing organic compounds, including CHOS and CHONS. As shown in **Fig. 1** and **Table S2**, the percentage of peaks response for S-containing species to total assigned peaks from vessels using HFO (35%) was higher than those from other vehicles, with 1, 3, 2 and 3 times more than those for excavators under idling, working, moving and vessels using diesel, respectively. However, the CHONS group for excavators was significantly higher than those for vessel emissions in terms of relative ions intensity. The high fraction of peak responses for S-containing

species from vessels using HFO might be attributed to the high sulfur content in HFO. The maximum sulfur content in HFO detected in this study was 2.46%, which was significantly higher than those in diesel (**Table 2**). In addition, for excavators under the idling mode, the fraction of the relative response of S-containing compounds was 32.5%, while for the working and moving modes, they were 11.6% and 17.1%, respectively.

To facilitate further discussion, three subgroups for CHONS ($N_1O_5S_1$, $N_1O_6S_1$ and $N_1O_7S_1$) and eight subgroups for CHOS ($O_3S_1$, $O_4S_1$, $O_5S_1$, $O_6S_1$, $O_6S_2$, $O_7S_1$, $O_7S_2$ and other) were characterized (**Fig. 4**). Generally, $O_4S_1$ and $O_5S_1$ were the most abundant subgroups for all off-road engines. For example, $N_1O_5S_1$ was the most abundant subgroup for the working mode, accounting for 36.7% of S-containing compounds. When $O/S \geqslant 4$, this indicates that a sulfate group exists within the organic compounds. Thus, $O_5S_1$ and $O_4S_1$ may be organosulfates or sulfonates (Riva et al., 2015). Riva et al. (2015) found that sulfur-containing products from PAHs were possible, and might not be solely sulfates but also sulfonates, especially with O/S values of 4-5. Upon comparing the sulfur-containing products observed in this study and the study by Riva et al., it was interesting to conclude that some PAH-derived OS products generated in the lab also have a significant response in field measurements. As shown as **Fig. S5**, three of the most abundant peaks ($C_8H_7O_5S^-$, $C_{11}H_5O_6S^-$ and $C_{18}H_{29}O_4S^-$) of S-containing compounds emitted from HFO-fueled vessels were also observed in the lab from PAH oxidations in the presence of sulfate. Although the fraction of S-containing compounds for vessels using HFO was similar to those for excavators under idling (**Fig.**

**1**), different structures of compounds might exist between these two engines. The relative response of $O_6S_2$ and $O_7S_2$ for excavators under idling was considerably higher than those for vessels using HFO.

S-containing compounds for vessels were highly unsaturated with 8.03 for the average DBE value which was higher than those for excavators (6.77; **Table S2**). Furthermore, the fraction of compounds with $Xn \geqslant 2.5$ accounted for 9.3%, 3.7% , 2.5%, 1.5% and 3.4% of the total S-containing compounds for vessels using HFO, diesel and excavators under idling, working and moving modes, respectively. Through a comparison of the average DBE value and fraction of compounds with $Xn \geqslant 2.5$ between excavators and vessels, it was proposed that different structures were present in S-containing compounds. The most abundant S-containing compounds emitted by off-road diesel engines were possibly aliphatic with long chains and sulfate fraction, which was consistent with the results of Tao et al. (2014) who found that most of the CHOS group contained long aliphatic carbon chains and low degrees of unsaturation and oxidation in ambient air in Shanghai. They suggested that most of these compounds were derived from diesel emission. It was interesting to infer that the most abundant peaks of CHOS compounds observed in this study were also identified through the laboratory simulation study (Riva et al., 2016). The conclusions reported from Riva's study could provide a possible chemical reaction path to explain the chemical formula detected from off-road engine combustion. The formulas marked in bold red in **Table S3** were the homologous compounds with $C_{12}H_{23}O_5S^-$ which was reported to have been generated from dodecane oxidation by Riva et al.'s research, while the formulas in bold

blue were likely formed from cycloalkanes. In contrast, the structures of S-containing compounds emitted from HFO fueled engines were liable to have condensed aromatic rings.

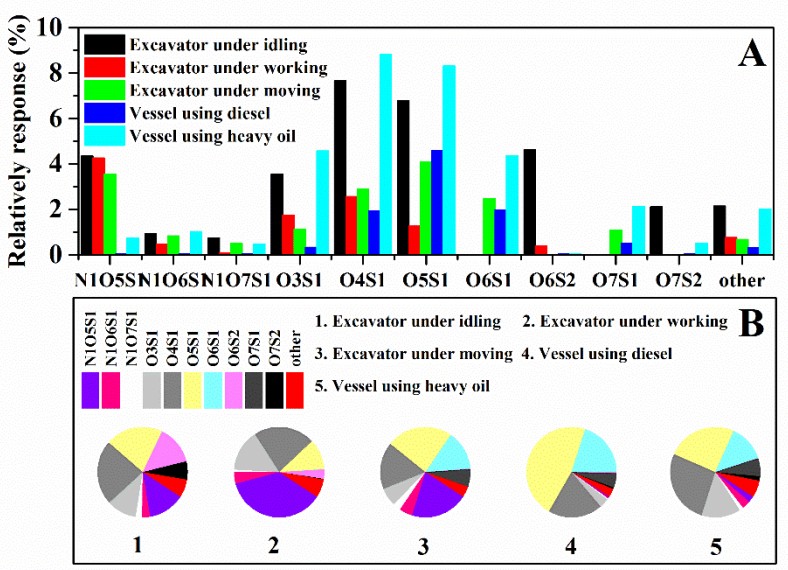

**Figure 4** The distribution of subgroups of S-containing compounds for off-road engines

On an average, 88.5% ± 9.1% and 98.1% ± 0.22% of CHOS compounds for vessels and excavators respectively were with O/S≥4, which possible to indicate that organic sulfates or sulfonates were the most impactful S-containing compounds emitted from off-road engines. For most CHOS compounds containing O/S≥4, O*(O*=O - 3) was used to identify the oxidation of organic compounds by removing the effect of sulfate or sulfonates. The modified VK diagram (H/C and O*/C) was used to characterize the structure of CHOS compounds (**Fig. 5**). Some organosulfates or sulfonates with an aromatic ring (subgroup A) could be produced by SOA precursors (e.g. α-pinene, β-pinene, monoterpenes) (Surratt et al., 2008). The most important precursors generating the subgroup A in this study might be the PAH-derived OS products in the presence of sulfate (Riva et al., 2015). However, there was another subgroup of organosulfates

(subgroup B) emitted by diesel vehicles with long aliphatic carbon chains (Tao et al., 2014). Subgroup B of organosulfates was defined as C > 8, DBE < 3, and 3 < O < 7 (red cycles in **Fig. 5**) (Tao et al., 2014), while the remaining CHOS compounds could be considered as subgroup A (blue cycle represented subgroup A with Xn ≥ 2.5 and the rest was contained within the black circle). The fractions of subgroup B of CHOS compounds in off-road diesel engine emissions (average: 33.9% ± 6.64%) were significantly higher than those in HFO fueled vessel emissions (19.9%). For vessels using HFO, almost 10% of CHOS compounds (blue in **Fig. 5**) were possibly organosulfates with one or more aromatic rings, which was consistent with the quality of the HFO. The structures detected for CHON compounds for HFO-fueled vessels and the organosulfates with one or more aromatic rings were similarly detectable in the atmosphere (Surratt et al., 2008).

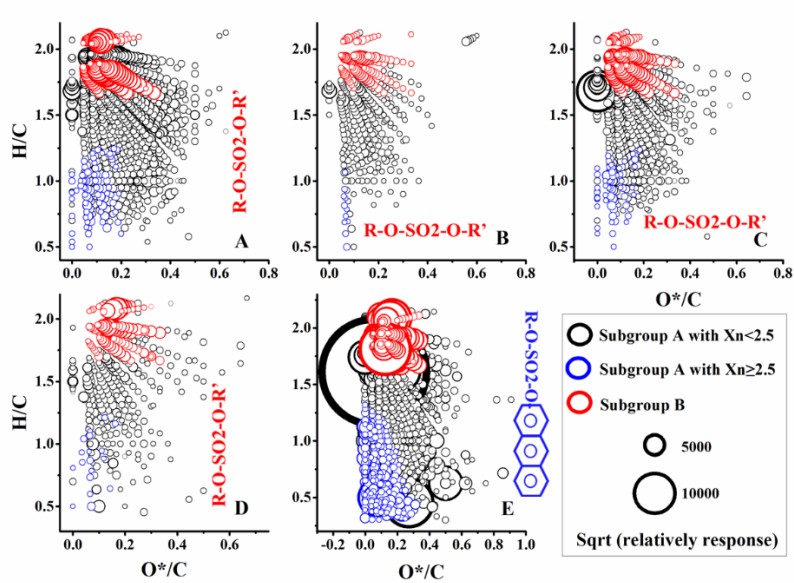

**Figure 5** The ratios of O*/C vs H/C of CHOS and proportion of subgroup B and Xn>2.5 for off-road engines (A, B, C, D and E were the mass spectra for excavators under the idling, working, and moving modes and vessels using diesel and HFO, respectively)

**3.5 Comparison of POCs in fresh smoke particles emitted from different sources**

CHO compounds were the most abundant species across all sources (biomass, coal, on-road vehicles and off-road vehicles) in terms of ions intensity, while the fractions of CHON and S-containing compounds were different from anthropogenic source emissions. Furthermore, the possible chemical structures of these compounds for diverse sources varied sharply.

For CHO compounds, the average DBE values from excavator and vessels emission were $8.38 \pm 3.84$ and $8.55 \pm 3.77$, which was higher than those emitted from crop burning (6.9) and coal combustion (7.48) (Song et al., 2018). The methoxyphenols ($C_9H_{14}O_4$, $C_{18}H_{28}O_8$, $C_{11}H_{20}O_5$, $C_{18}H_{22}O_7$, $C_9H_{12}O_4$) and fatty acids ($C_{16}H_{32}O_2$), derived from limonene and palmitic acid, were frequently observed in crop burning emissions. CHO compounds with high C numbers ($C_{20}H_{28}O_2$, $C_{20}H_{26}O_2$, $C_{20}H_{30}O_2$) were detected from wood burning. One benzene ring substituted with O-containing groups such as hydroxyl, methoxyl, and carboxyl ($C_8H_6O_4$, $C_{13}H_{18}O_4$ and $C_{12}H_{16}O_4$) were dominant in coal combustion. Overall, mono-aromatics dominated tunnel samples (Tong et al., 2016) and off-road diesel vessels. However, abundant condensed aromatic ring structures with high O numbers ($C_{10}H_5O_8$, $C_9H_5O_6$ and $C_{10}H_5O_9$) were proposed for HFO fueled vessels.

For CHON compounds, almost all sources were reported to emit nitrophenol compounds, while the substituted groups were slightly different due to different numbers of N and O atoms. The fraction of relative peak response of CHON compounds, an important light-absorbing substance, could reach half of the POCs from biomass

burning emission. Methyl-nitrocatechols produced from the oxidation of cresol and N-bases composed of C, H, and N elements were considered as the biomarkers for biomass burning (Laskin et al., 2009; Wang et al., 2017). However, on comparing the signal intensity of nitroaromatics in ambient aerosol and fresh biomass burning smoke, Wang et al. (2017) found evidence to the contrary. Signal intensity was stronger in ambient aerosols than that in fresh biomass burning smoke, which indicated the existence of other sources or aging process. Recently, nitrophenol was also detected in tunnel samples indicating traffic sources. In this study, we inferred that dinitrophenol were abundant in off-road diesel vehicle emissions, while nitronaphthol with one or more methyl groups was dominant for HFO-fueled vessels emission.

Except for biomass burning, S-containing compounds were still an important group of organic matter for coal combustion, on-road vehicles, off-road diesel vehicles, HFO-fueled vessels, and in background sites, accounting for 48%, 17%, 8.9%, 33% and 32%, respectively, of total detected organic matters. Organosulfates and sulfonates were one of the most important HULIS, which were reported as the prominent S-containing compounds at background sites due to aging reactions of organics with $H_2SO_4$. For coal combustion, S-containing compounds had low DBE and $AI_{mod,w}$ values, which were probably considered as alkylbenzene rings substituted with one sulfate group. The specific structure of S-containing compounds was organosulfates or solfonates with condensed aromatic rings for HFO-fueled vessels, while more abundant organosulfates with aliphatic chains were observed in emissions from off-road diesel equipment. Likewise, the organosulfates with aliphatic long chains alkanes were observed in on-

road traffic emissions due to its original oil structure (Jiang et al., 2016; Riva et al., 2016; Tao et al., 2014). These S-containing compounds with high aromaticity, or long chain alkanes, were frequently formed from secondary photochemical reactions between oxidation products of volatile and intermediate volatility organic compounds

and acidified sulfate particle (Riva et al., 2015; 2016). Therefore, a high-abundance of S-containing compounds in the atmosphere might from secondary photochemical reactions and also be emitted directly from the combustion of off-road engines.

## 4 Atmospheric implications

In recent years, air pollution from ship emissions, especially in coastal areas, has

10 drawn increasing worldwide attention. In previous studies, vanadium (V) and nickel (Ni) were widely used as specific tracers for heavy fuel oil (HFO) fueled vessel emissions (Liu et al., 2017b). It was reported that the V content in HFO gradually decreased from 39.5 ppm in 2013 to 12.7 ppm in 2018 in China in compliance with the requirements for all vessels within China domestic emission control areas (DECA)

(Zhang et al., 2019). In addition, as V and Ni could also originate from industrial emissions, the uncertainties in estimating ship emission contributions to atmospheric $PM_{2.5}$ could increase if only these heavy metals are used as tracers (Zhang et al., 2014). Meanwhile, the relative intensity of OC components from HFO-fueled vessel emissions increased from 10.9% to 21.7% after the implementation of DECA policy (Zhang et al.,

2019), while the sulfur content of some heavy oils remained higher than those regulated by oil standards of ships (Zhang et al., 2018). Therefore, probing the molecular-level characteristics of S-rich organic matter from HFO-fueled vessel emissions will be

effective and useful.

In this study, we found that the molecular compositions of POCs emitted by HFO-fueled vessels were different from those of other source emissions (e.g. off-road diesel engines, biomass and coal burning). The results of this study demonstrated that soot-materials or oxidized polycyclic aromatic hydrocarbons and S-containing species especially those with high aromaticity and O/S ratios most likely organosulfates or sulfonates constitute significant fractions of POCs in HFO-fueled vessel smoke, and were significantly different from those in other primary source emissions. Therefore, it provides an opportunity to act as potential molecular markers to distinguish HFO-fueled vessel emissions in the future.

On the other hand, it should be noted that these organosulfates and sulfonates have been previously reported to be mainly formed from secondary photochemical reactions via the oxidation products of VOCs and acidified sulfate seed particles or sulfuric acid in atmosphere (Riva et al., 2015; Tao et al., 2014). However, the high abundance of S-containing species found in HFO-fueled vessel smoke indicates that not only secondary organic aerosols but also primary HFO-fueled vessel smoke could be an important source of organosulfates (**Figure S5**). Therefore, ignoring the contributions of HFO-fueled vessel emissions to organosulfates might lead to the overestimation of the contribution of secondary organic aerosols in the atmosphere.

Although some useful characterizations of POCs from off-road engine combustion emissions were proposed, some issues still need to be resolved in the future. This include, (1) determination of the molecular structure of the distinctive compounds

mentioned in this study should be further explored; and (2) potentially different molecular structures of organosulfates from HFO-fueled vessel emissions and SOA should also be distinguished.

**Author contribution.** MC and CL contributed equally to this work. MC wrote the manuscript in close cooperation with CL and got helpful direction by YC, JL (Jun Li) and JZ. FZ, JL (Jia Li) and YM were responsible for sampling and chemical analysis. BJ, CY and MZ were familiar with data process of FT-ICR MS and mass absorption efficiency. ZX and GZ provided key contributions to article structure and logic.

**Acknowledgements.** This study was supported by the Natural Scientific Foundations of China (Nos. 91744203 and 41773120), Guangdong Provincial Science and Technology Planning Project of China (No. 2017B050504002) and State Key Laboratory of Organic Geochemistry, Guangzhou Institute of Geochemistry (No. SKLOG-201732).

**Competing interests.** The authors declare that they have no conflict of interest.

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
