# Peer review of "Molecular Characterization of Polar Organic Matters in Off-road Engine Emissions Using Fourier Transform Ion Cyclotron Resonance Mass Spectrometry (FT-ICR MS): New Direction to Find Biomarkers"

_Atmospheric Chemistry and Physics, 2019_

## Referee Comment (RC1) · Anonymous Referee #1 · 17 Jun 2019

Overall comments: The study focused on the elemental compositions of particulate matters emitted from the typical off-road engines in China. The chemical structures and molecular information of major emissions are inferred in some cases, which should potentially benefit the research community in terms of the tracers indicating off-road engine emissions. Differences were clearly demonstrated between the emissions from diesel engines and those fueled by heavy fuel oils, particularly the degree of unsaturation and oxidation. While the paper will likely be a significant contribution, revisions and

clarifications are needed at this moment. Firstly, the English should be substantially improved before I can comprehensively evaluate the quality and value of the paper. The language and writing skills in the present manuscript seriously hinder the transferring of knowledge to the readers, as well as the objective evaluation on the work. I would recommend the manuscript to be edited by an editing company. Then, the significance of the paper should be more clearly demonstrated. The knowledge gaps illustrated in the introduction include (1) the unavailability of unique tracers for separating the on-road and off-road engine emissions; and (2) the challenge in detecting the large molecule and polar markers by the conventional GC-MS. However, I do not think the study filled the gaps sufficiently. How will the elemental compositions of off-road engine emissions contribute to the identification of these emissions in airborne particulate matters? Obviously, the molecular information of the tracers inferred in this study is inadequate. How about the differences in elemental compositions between the on-road and off-road engine emissions? Can the findings in this study be directly used in the concurrent source apportionment techniques, such as the filter based and AMS based source apportionment? In other words, there should be a section in this paper showing the implications of the study.

Specific comments: 1. Improve the English throughout the manuscript. Lines 13-15, page 3, line 1, page 4, lines 10-13, page 4. Too many grammatical errors, and I cannot list of them here. 2. Methodology: How many samples were collected in total and in each scenario? How did you combine the samples? What was the purpose of combining the samples given the expected enough loading of PM for chemical analysis? How to consider the variations among the samples collected in the same scenarios? The representativeness of the samples should be discussed. 3. Off-road and non-road are alternatively used. Keep consistent throughout. 4. Lines 5-8, page 12: Do you mean the number of peaks for CHO compounds? The similarly inaccurate descriptions appeared many times in the manuscript, which need to be double checked and clarified. 5. Lines 8-12, page 12: What are the ranges of number of peaks detected in biomass and coal combustions, and the references? 6. Lines 1-5, page 14: References must be

provided to support the interpretations on the regions in Figure 2. 7. Lines 9-13, page 16: How does kinematical viscosity account for the high oxygen content in HFO-fueled vessel emissions? 8. Lines 4-6, page 17. I do not get the point why the discussions only focus on the excavators under the working mode and vessels using HFO, given that the CHON structures were different even among the excavators as mentioned in the previous sentence. 9. Lines 8-20, page 17: The inferences on the structures of CHON must be illustrated in more details. Was any of the proposed compounds ever reported in previous studies? 10. Figure 3: I do not think the structures of dinitrophenol and methyl dinitrophenol have been correctly presented in Figure 3, same for nitron-aphthol and methyl nitronaphthol. Please clarify. 11. Lines 1-6, page 20. What are the bases that the conclusions can be drawn? For example, "The most of S-containing compounds emitted from off-road diesel engines were aliphatic with long chains and sulfate fraction". 12. Lines 9-11, page 20: Descriptions should be more accurate. I suppose you mean that organic sulfates were the most important S-containing compounds emitted from off-road engines. 13. Line 12, page 20: This expression "O*(O-3)" will mislead the readers. Change it to O* (O* = O - 3). 14. As mentioned earlier, the implications of this study should be summarized and clearly demonstrated, rather than a simple summary of the findings. 15. English and writing skills must be substantially improved. Otherwise, it will be impossible for this paper to be published on ACP.

---

## Referee Comment (RC2) · Anonymous Referee #2 · 31 Jul 2019

Summary and Recommendation:

This study applied FR-ICR MS equipped with electrospray ionization (ESI) to chemically characterize polar organic compounds emitted from off-road engine combustion sources. The authors generated these emissions from engine types that appear to be relevant to China. The intentions of this study are very good, and will certainly be of interest to the readership of Atmospheric Chemistry and Physics. However, there

are many shortcomings with the present manuscript that I will outline below in detail with my specific comments. In short, there are two major issues with this manuscript: (1) although the authors are likely not all native-English speakers, I found many portions of the text hard to follow or even distracting due to the poor English grammar or misspellings. In my technical (minor) comments below I outline some of them, but I don't feel the burden should be on reviewers to correct all of these mistakes; and (2) the chemical method (or approach) used to chemically characterize the polar organic aerosol constituents is flawed in how it was operated and this needs to be thoroughly addressed before I can recommend this manuscript for publication in ACP. Due to the nature of my specific comments below, I must recommend this manuscript be reconsidered after major revisions.

Specific Comments:

1.) Lack of Chromatographic Separation Before ESI-MS Detection:

My biggest concern with this study is the lack of chromatographic separation before ESI-FT-ICR MS detection. Since chromatographic separation was not used, the authors were forced to utilize SPE to desalt the filter samples. The reason for this desalting step is that inorganic ions can cause unwanted adduct formations and ion suppression effects during ESI-MS analyses, both of which can lead to a misinterpretation of the "actual" chemical composition of polar organic aerosol constituents. Without chromatographic separation, such as reverse-phase liquid chromatography (RPLC) or hydrophilic interaction liquid chromatography (HILIC), it is difficult to resolve isomers from each other and also due to ion suppression/matrix effects that result from direct infusion (which was done here) quantitating is near impossible. Thus, the authors are forced to only report molecular formulas. So the qualitative results reported here is thus how many number of ions were detected with CHO, CHON, CHOS, and CHONS. Just because you may have a large number of a certain type of chemical class, doesn't necessarily translate into abundance of polar organic material. The authors imply that simply having ultra-high mass resolution is enough to justify the results from this study.

I would argue this is only true if the complex organic matrix is chromatographically separated online before ESI-MS detection. ESI-MS is notoriously known to have major issues if this is not done.

Finally, one major issue with the SPE method as described here is you severely risk removing the most polar and water-soluble organic compounds that can't be retained by the Oasis HLB SPE cartridge. Previous field samples analyzed by both Gao et al. (JGR) and Surratt et al. (2007, ES&T) from the Seinfeld group at Caltech revealed that SPE caused isoprene-derived SOA constituents, which are very polar and water-soluble, to be completely removed during SPE treatment. As a result, the isoprene SOA constituents were not reported in Gao et al. (2006, JGR).

At minimum, the authors need to address these limitations either in the experimental method and/or in the results and discussion section.

2.) Experimental Section, Filter Extraction Method:

Were quality control tests conducted to ensure that organic aerosol constituents were effectively removed from the filter media during filter extraction? If not, this should likely be done and reported in a revised manuscript. Also, how much negative artifacts (or losses) do you expect occur during your filter extraction process? Also, by using water to extract the filters by sonication, do you worry that oxidants (e.g., OH radicals) are produced that can degrade your aerosol constituents or even transform them into unintended products?

3.) Experimental, Page 8, Lines 9-18:

How many quartz filters were combined for extraction for chemical analyses? Was it 5 filters collected from the same vessel and operating condition?

4.) Experimental, Filter Collection Details:

What was the flowrate used for PM2.5 sampling? Did you have denuders? If not, what potential positive artifacts occurred on your quartz filters when conducting the

molecular composition analyses? Don't you expect some absorption of semivolatiles on these filters?

5.) I would consider changing title to:

Molecular Characterization of Polar Organic Aerosol Constituents in Off-Road Engine Emissions Using Fourier Transform Ion Cyclotron Resonance Mass Spectrometry (FT-ICR MS): Implications for Source Apportionment

6.) Page 19, Line 10:

The authors might want to look at Riva et al. (2015, ES&T) from the Surratt group. They found that sulfur-containing products from PAHs were possible, and may not be solely sulfates but also sulfonates, especially with O/S values of 4-5. It would be interesting to know if you observed any of these PAH-derived OS products that they generated in the lab from PAH oxidations in the presence of sulfate.

7.) Page 20, Lines 1-6:

Are the authors also familiar from work published by Riva et al. (2016, ACP) from the Surratt group on organosulfates from the oxidation of long-chain alkanes. It would interesting to know if you observed similar molecular formulas to that study.

Minor (technical) Comments:

1.) Abstract: The last sentence of the abstract needs to be completely re-worded. The current sentence is poorly worded and not easy to understand.

2.) I would change "polar organic matters (POM)" to polar organic compounds (POCs).

3.) Introduction, Page 6, Lines 6-8:

Change this sentence to state:

"This study aimed to chemical characterize polar organic aerosol constituents at the molecular level that are emitted from typical non-road engines by FT-ICR MS to provide

Interactive
comment
new aerosol marker compounds for non-road engines."

4.) Page 7, Line 5: Do you mean to say "plume" instead of "flume" ?

5.) Experimental Section, Page 8, Line 14: Delete "continually"
* * *

---

## Author Comment (AC1) · 6 Sep 2019

The comment was uploaded in the form of a supplement:
https://www.atmos-chem-phys-discuss.net/acp-2019-449/acp-2019-449-AC1-
supplement.zip

2019.

---

## Author Comment (AC2) · 6 Sep 2019

The comment was uploaded in the form of a supplement:
https://www.atmos-chem-phys-discuss.net/acp-2019-449/acp-2019-449-AC2-supplement.zip

---

## Author Response (AR1)

**Response to the Reviewers' comments**

**Journal:** Atmospheric Chemistry and Physics

**Manuscript ID:** acp-2019-449-RC1

**Title:** Molecular Characterization of Polar Organic Aerosol Constituents in Off-Road Engine Emissions Using Fourier Transform Ion Cyclotron Resonance Mass Spectrometry (FTI-CR MS): Implications for Source Apportionment

**Author(s):** Min Cui, Cheng Li, Yingjun Chen*, Fan Zhang, Jun Li*, Bin Jiang, Yangzhi Mo, Jia Li, Caiqing Yan, Mei Zheng, Zhiyong Xie, Gan Zhang, and Junyu Zheng*

**Corresponding author:** Yingjun Chen; Jun Li and Junyu Zheng (email: yjchentj@tongji.edu.cn; junli@gig.ac.cn; zhengjunyu_work@hotmail.com)

Thank you so much for your consideration! Also, the anonymous reviewer's comments are highly appreciated! So far, we have revised the manuscript accordingly. Our point-by-point responses (in black) to each reviewer's comments are listed below. And the modifications in the revised manuscript with marks are marked in blue. Please see the manuscript for details.

**Response to reviewer's 1**

***Comment #1****:* Firstly, the English should be substantially improved before I can comprehensively evaluate the quality and value of the paper. The ACPD language and writing skills in the present manuscript seriously hinder the transferring of knowledge to the readers, as well as the objective evaluation on the work. I would recommend the manuscript to be edited by an editing company.

**Response:** Thanks. A professional language editing company has thoroughly polished and edited the revised manuscript (**Fig. R1**).

**Figure R1** Certificate of english editing

***Comment #2****:* The knowledge gaps illustrated comment in the introduction include (1) the unavailability of unique tracers for separating the on-road and off-road engine emissions; and (2) the challenge in detecting the large molecule and polar markers by the conventional GC-MS. However, I do not think the study filled the gaps sufficiently. How will the elemental compositions of off-road engine emissions contribute to the identification of these emissions in airborne particulate matters? Obviously, the molecular information of the tracers inferred in this study is inadequate. How about the differences in elemental compositions between the on-road and off-road engine emissions? Can the findings in this study be directly used in the concurrent source apportionment techniques, such as the filter based and AMS based source

**Response:** We appreciate the reviewer's constructive suggestion. We recognize that the knowledge gaps illustrated in the introduction might not necessarily be filled with the results presented. Thus, some modifications and discussions were added in introduction and implications in the revised manuscript (**Page 4 lines 13-15; Page 5 lines 20-22; Page 6 lines 1-8; Page 23 lines 1-6; Page 27 lines 1-22; Page 28 lines 1-22; Page 29 lines 1-4**).

The purpose of this research was divided into two aspects in the revised introduction. One was to investigate the characterization of polar organic constituents at the molecular level to serve as molecular markers from off-road engines (**Page 4 lines 13-15**), which was sparsely reported in previous research. Meanwhile, the differences between the chemical characterization of polar organic matters (POCs) emitted from various sources were discussed in the revised implications (**Page 27 lines 1-22; Page 28 lines 1-22; Page 29 lines 1-4).** Moreover, we found that the organosulfates or sulfonates with condensed aromatic rings could be a unique tracer for heavy-oil fueled vessel emissions.

The other objective was to speculate the possible chemical structure of N-containing and S-containing organic compounds. These are considered one of the most important secondary organic aerosols (SOA) and HUmic-LIke Substances (HULIS) in the atmosphere to provide useful information to identify the significant role of off-road engine combustion in climate change or SOA formation (**Page 5 lines 20-22; Page 6 lines 1-8**). By comparing the sulfur-containing products observed in this study and the chamber experiment, it was interesting to have found that some PAH-derived OS products generated in the lab also have significant response in field measurements (**Page 23 lines 1-6**). We concluded that a high abundance of S-containing compounds in the atmosphere might come from secondary photochemical reactions as well as directly from the combustion of off-road engines.

***Comment #3****:* Improve the English throughout the manuscript. Lines 13-15, page 3, line 1, page 4, lines 10-13, page 4. Too many grammatical errors, and I cannot list of them here.

**Response:** Thanks. A professional language editing company has thoroughly polished and edited the revised manuscript.

***Comment #4****:* Methodology: How many samples were collected in total and in each scenario? How did you combine the samples? What was the purpose of combining the samples given the expected enough loading of PM for chemical analysis? How to consider the variations among the samples collected in the same scenarios? The representativeness of the samples should be discussed.

**Response:** As shown in **Table S1**, we selected four excavators, two diesel-fueled vessels, and two HFO-fueled vessels. For each excavator, we sampled from three operation modes. There were two important reasons to combine the samples. First, to get enough loading of organic matter. The second was that the analysis of FT-ICR MS samples was cost-prohibitive. We believed that combining samples from the same types of vehicles on the same operation modes could remove the random error, which could better represent the average emission status (**Page 10 lines 6-9**).

***Comment #5****:* Off-road and non-road are alternatively used. Keep consistent throughout.

**Response:** Thank you. We have used "off-road" throughout the revised manuscript.

***Comment #6****:* Lines 5-8, page 12: Do you mean Printer-friendly version the number of peaks for CHO compounds? The similarly inaccurate descriptions appeared many times in the manuscript, which need to be double checked and clarified.

**Response:** Thank you. We have checked the inaccurate descriptions, and modified them throughout the revised manuscript.

***Comment #7****:* Lines 8-12, page 12: What are the ranges of number of peaks detected in biomass and coal combustions, and the references?

**Response:** Thank you. We have added the ranges of the number of peaks detected in biomass and coal in the revised manuscript, along with the references (**Page 15 lines 6-7**).

***Comment #8***: Lines 1-5, page 14: References must be provided to support the interpretations on the regions in Figure 2. 7. Lines 9-13, page 16: How does kinematical viscosity account for the high oxygen content in HFO-fueled vessel emissions?

**Response:** References have been provided to support the interpretations of the regions in **Fig. 2** in the revised manuscript (**Page 16 lines 13-16**). It was reported that the atomization of fuel drops was strongly attributed to their kinematic viscosity (Örs et al., 2018). High viscosity lead to poor atomization, which could cause incomplete combustion and result in an increased number of oxygen attachments onto the organic matter. Furthermore, high viscosity always accompanied high fatty acid compounds containing higher oxygen contents (Ramos et al., 2009).

"As shown in **Fig. 2**, region 1 represented monocarboxylic acid, which was more abundant in both idling and moving modes than in the working mode (Wozniak et al., 2008; Lin et al., 2012). Region 2 represented compounds with low ratios of H/C and O/C and DBE>10 which were commonly considered as condensed hydrocarbons (Wozniak et al., 2008; Lin et al., 2012)."

**References:**

[1] Wozniak, A. S., Bauer, J. E., Sleighter, R. L., Dickhut, R. M. and Hatcher, P. G.: Technical Note: Molecular characterization of aerosol-derived water-soluble organic carbon using ultrahigh resolution electrospray ionization Fourier transform ion cyclotron resonance mass spectrometry, Atmospheric Chemistry and Physics, 8 (17): 5099-5111, 2008.

[2] Lin, P., Rincon, A. G., Kalberer, M. and Yu, J. Z.: Elemental Composition of HULIS in the Pearl River Delta Region, China: Results Inferred from Positive and Negative Electrospray High Resolution Mass Spectrometric Data, Environmental Science & Technology, 46 (14): 7454-7462, 2012.

[3] Örs, I., Sarıkoç, S., Atabani, A. E., Ünalan, S. and Akansu, S. O.: The effects on performance, combustion and emission characteristics of DICI engine fuelled with $TiO_2$ nanoparticles addition in diesel/biodiesel/n-butanol blends, Fuel, 234: 177-188, 2018.

[4] Ramos, M. J., C. M. Fernandez, A. Casas, L. Rodriguez and A. Perez. Influence of fatty acid composition of raw materials on biodiesel properties. Bioresour Technol 100(1): 261-268, 2009.

***Comment #9***: Lines 4-6, page 17. I do not get the point why the discussions only focus on the excavators under the working mode and vessels using HFO, given that the CHON structures were different even among the excavators as mentioned in Interactive the previous sentence.

**Response:** We are thankful for the reviewer's kind suggestion. The main structures of the CHON group from excavators under three operation modes and vessels using HFO and diesel were illustrated in **Fig. 3** in the revised manuscript. Although the fractions of the CHON group for excavators under different operation modes were different, the probable chemical structures were exactly the same, as shown in Figure 3. The meaning of the referred sentence might seem incorrect. Thus, the sentence has been deleted, and the structures of all the off-road engines were discussed in the revised manuscript (**Page 21 Figure 3**).

[Figure]

**Figure 3** Molecular composition and possible structure of CHON for excavators under three

modes and vessels using HFO and diesel

***Comment #10***: Lines 8-20, page 17: The inferences on the structures of CHON must be illustrated in more details. Was any of the proposed compounds ever reported in previous studies?

**Response:** Thank you for pointing this out. Detailed inferences on the structures of CHON were illustrated in the revised manuscript (**Page 19 lines 21-22; Page 20 lines 1-9**). The proposed compounds were also reported in previous research (Yassine et al., 2014; Tong et al., 2016).

"As mentioned, the most abundant relative responses of CHON group for diesel-fueled excavators and vessels were $C_{10}H_5N_2O_5$, $C_{11}H_7N_2O_5$, and $C12H9N2O5$, which compose the largest green ball in **Fig. 3** with Xn=2.5, indicating the presence of a benzene core structure in the compounds. Thus, it was most likely dinitrophenol, and methyl dinitrophenol compounds. Likewise, $C_{10}H_4NO_6$, $C_9H_4NO_4$ and $C_{10}H_4NO_7$

comprise the largest yellow ball in Fig. 3 for HFO-fueled vessels, most of which have Xn>2.7 indicating the presence of condensed aromatic compounds. Nitronaphthol and methyl nitronaphthol were the most significant compounds arising from HFO-fueled vessel emissions, which have previously been reported in vehicle emissions (Yassine et al., 2014; Tong et al., 2016)."

**References:**

[1] Yassine, M. M., Harir, M., Dabek-Zlotorzynska, E. and Schmitt-Kopplin, P.: Structural characterization of organic aerosol using Fourier transform ion cyclotron resonance mass spectrometry: Aromaticity equivalent approach, Rapid Communications in Mass Spectrometry, 28 (22): 2445-2454, 2014.

[2] Tong, H. J., Kourtchev, I., Pant, P., Keyte, I. J., O'Connor, I. P., Wenger, J. C., Pope, F. D., Harrison, R. M. and Kalberer, M.: Molecular composition of organic aerosols at urban background and road tunnel sites using ultra-high resolution mass spectrometry, Faraday Discussions, 189: 51-68, 2016.

***Comment #11****:* Figure 3: I do not think the structures of dinitrophenol and methyl dinitrophenol have been correctly presented in Figure 3, same for nitronaphthol and methyl nitronaphthol. Please clarify.

**Response:** Thanks. The correct structures have been redrawn in **Fig. 3** in the revised manuscript (**Page 21 Figure 3**).

***Comment #12****:* Lines 1-6, page 20. What are the bases that the conclusions can be drawn? For example, "The most of S-containing compounds emitted from off-road diesel engines were aliphatic with long chains and sulfate fraction".

**Response:** Thanks for your suggestion. The bases have been drawn in **Fig. 5** in the revised manuscript (**Page 26 Figure 5**).

[Figure]

**Figure 5** The ratios of O*/C vs H/C of CHOS and proportion of subgroup B and Xn>2.5 for off-road engines (A, B, C, D and E were the mass spectra for excavators under the idling, working, moving and vessels using diesel and HFO, respectively)

*Comment #13*: Lines 9-11, page 20: Descriptions should be more accurate. I suppose you mean that organic sulfates were the most important S-containing compounds emitted from off-road engines.

**Response:** Thanks for your kindly suggestion. The sentence has been modified in the revised manuscript (**Page 24 lines 14-16**).

"On an average, 88.5%±9.1% and 98.1%±0.22% of CHOS compounds for vessels and excavators respectively were with O/S⩾4, which indicated that organic sulfates or sulfonates were the most impactful S-containing compounds emitted from off-road engines."

**_Comment #14_**: Line 12, page 20: This expression "$O*(O-3)$" will mislead the readers. Change it to $O*$ ($O* = O - 3$).

**Response:** Thanks for reminding. The suitable expression has been changed in the revised manuscript (**Page 25 line 1**).

**_Comment #15_**: As mentioned earlier, the implications of this study should be summarized and clearly demonstrated, rather than a simple summary of the findings.

**Response:** Thank you for your kindly suggestion. The implication of this study has been summarized in the revised manuscript (**Page 27 lines 1-22; Page 28 lines 1-22; Page 29 lines 1-4**).

**_Comment #16_**: English and writing skills must be substantially improved. Otherwise, it will be impossible for this paper to be published on ACP.

**Response:** Thanks. A professional language editing company has thoroughly polished and edited the revised manuscript.

**Response to reviewer's 2**

**Comment #1**: *Although the authors are likely not all native-English speakers, I found many portions of the text hard to follow or even distracting due to the poor English grammar or misspellings. In my technical (minor) comments below I outline some of them, but I don't feel the burden should be on reviewers to correct all of these mistakes.*

**Response:** Thanks. A professional language editing company has thoroughly polished and edited the revised manuscript (**Fig. R1**).

[Figure]

**Figure R1** Certificate of english editing

**Comment#2:** Lack of Chromatographic Separation Before ESI-MS Detection:

My biggest concern with this study is the lack of chromatographic separation before

ESI-FT-ICR MS detection. Since chromatographic separation was not used, the authors

were forced to utilize SPE to desalt the filter samples. The reason for this desalting step is that inorganic ions can cause unwanted adduct formations and ion suppression effects during ESI-MS analyses, both of which can lead to a misinterpretation of the "actual" chemical composition of polar organic aerosol constituents. Without chromatographic separation, such as reverse-phase liquid chromatography (RPLC) or hydrophilic interaction liquid chromatography (HILIC), it is difficult to resolve isomers from each other and also due to ion suppression/matrix effects that result from direct infusion (which was done here) quantitating is near impossible. Thus, the authors are forced to only report molecular formulas. So the qualitative results reported here is thus how many number of ions were detected with CHO, CHON, CHOS, and CHONS. Just because you may have a large number of a certain type of chemical class, doesn't necessarily translate into abundance of polar organic material. The authors imply that simply having ultra-high mass resolution is enough to justify the results from this study.

 I would argue this is only true if the complex organic matrix is chromatographically separated online before ESI-MS detection. ESI-MS is notoriously known to have major issues if this is not done.

Finally, one major issue with the SPE method as described here is you severely risk removing the most polar and water-soluble organic compounds that can't be retained by the Oasis HLB SPE cartridge. Previous field samples analyzed by both Gao et al. (JGR) and Surratt et al. (2007, ES&T) from the Seinfeld group at Caltech revealed that SPE caused isoprene-derived SOA constituents, which are very polar and watersoluble, to be completely removed during SPE treatment. As a result, the isoprene SOA constituents were not reported in Gao et al. (2006, JGR).

At minimum, the authors need to address these limitations either in the experimental method and/or in the results and discussion section.

**Response:** Thanks for your constructive suggestion. We are completely in agreement with the reviewer's opinion that FT-ICR MS without chromatographic separation fails to recognize the isomers of POCs, and that an appropriate description of this limitation should be mentioned in the revised manuscript (**Page 6 lines 15-20**) to avoid misleading

readers into thinking that this method is infallible. However, due to the high resolving power of FT-ICR MS, it is widely and successfully used to explore the chemical compositions of macromolecular polar organic compounds. Therefore, the chemical compositions and structures of POCs emitted from off-road engines were detected and deduced in this study by using FT-ICR MS and some empirical values.

Furthermore, it was reported that SPE methods for desalting could also remove a majority of the inorganic ions and low molecular weight organic compounds, such as some isoprene derived organosulfates and sugars. Thus, this limitation was also addressed in the revised manuscript (**Page 11 lines 8-11**).

"It should be noted that FT-ICR MS, without chromatographic separation, can only detect molecular formulas and molecular identification based on elemental composition alone. This is challenging because most complex molecules have several stable isomeric forms."

"A majority of inorganic ions (e.g. ammonium, sulfate, and nitrate) and low-molecular-weight organic compounds such as isoprene-derived organosulfates and sugars could be removed during SPE treatment (Gao et al. 2006, Lin et al. 2012, Surratt et al. 2007), which were not discussed in this research."

**Table R1** Recovery efficiencies for known organic compounds (from Fan et al., 2012)

| Tested substances | ENVI-18 | | HLB | | XAD-8 | | DEAE | |
|---|---|---|---|---|---|---|---|---|
| | TOC | UV (250 nm) | TOC | UV (250 nm) | TOC | UV (250 nm) | TOC | UV (250 nm) |
| **Carbonyls** | | | | | | | | |
| Glyoxal | $3.7 \pm 1.0^a$ | na[b] | $7.6 \pm 0.7$ | na | nd[c] | na | $4.0 \pm 0.3$ | na |
| **Monocarboxylic acid** | | | | | | | | |
| Acetic acid | $2.8 \pm 0.1$ | na | $4.7 \pm 1.1$ | na | nd | na | nd | na |
| ʟ-Lactic acid | $29.4 \pm 5.4$ | na | $32.2 \pm 1.6$ | na | $9.92 \pm 0.9$ | na | nd | na |
| **Dicarboxylic acid** | | | | | | | | |
| Succinic Acid | $7.5 \pm 0.5$ | na | $33.8 \pm 0.3$ | na | $25.3 \pm 4.1$ | na | nd | na |
| Suberic Acid | $91.3 \pm 0.3$ | na | $95.0 \pm 1.0$ | na | $75.0 \pm 19.6$ | na | $15.6 \pm 1.3$ | na |
| **Aromatic acid** | | | | | | | | |
| 3,5-Dihydroxybenzoic | $70.4 \pm 2.1$ | na | $91.7 \pm 1.6$ | na | $72.9 \pm 6.4$ | na | $98.5 \pm 1.5$ | na |
| Phthalic Acid | $89.4 \pm 0.6$ | na | $98.6 \pm 6.2$ | na | $97.7 \pm 2.0$ | na | $18.1 \pm 0.3$ | na |
| **Phenols** | | | | | | | | |
| Guaiacol | $3.0 \pm 0.0$ | na | $4.3 \pm 0.0$ | na | nd | na | $2.4 \pm 0.2$ | na |
| 4′-Hydroxyacetophenone | $87.5 \pm 1.5$ | na | $95.2 \pm 1.1$ | na | $0.33 \pm 0.0$ | na | $2.7 \pm 0.4$ | na |
| **Saccharides** | | | | | | | | |
| Sucrose | $1.8 \pm 1.0$ | na | $4.7 \pm 0.9$ | na | nd | na | $0.7 \pm 0.1$ | na |
| **Humic substances** | | | | | | | | |
| Suwannee river fulvic acid (SRFA) | $94.2 \pm 0.3$ | $98.6 \pm 0.3$ | $91.4 \pm 1.7$ | $92.5 \pm 1.7$ | $94.3 \pm 8.8$ | $94.2 \pm 6.1$ | $94.4 \pm 0.6$ | $96.7 \pm 0.6$ |
| Pohakee peat humic acid (PPHA) | $39.3 \pm 2.5$ | $46.7 \pm 0.1$ | $29.2 \pm 0.7$ | $32.9 \pm 0.3$ | $61.7 \pm 3.6$ | $59.3 \pm 2.8$ | $44.4 \pm 0.8$ | $56.2 \pm 1.0$ |

[a] Standard deviations were obtained based on a series of triplicate trials.
[b] Not analysis.
[c] Not detectable.

(3) Extraction by pure water was a common method to detect water-soluble POCs (Song et al., 2018; Wang et al., 2017). It was reported that only ultraviolet irradiation, electrolysis, or heating could promote OH radical formation and reaction with some organic matters (Li et al., 2019; Staudt et al., 2014). Ice bags were used throughout the ultrasound process to reduce the temperature, and we

believed that this way, certain oxidation products cannot be formed.

"Due to the limitations of organic matter load in filters and cost-prohibitive analysis, the filters sampled from off-road engines with the same operation modes or fuel quality

were combined together to characterize the comprehensive molecular compositions of POCs for off-road engines under different operation modes and fuel quality. As shown in **Table S1**, five samples (1, 2, 3, 4 and 5) were selected to conduct FT-ICR MS analysis, which represented vessels using heavy fuel oil, vessels using diesel, excavators under idling, moving, and working modes, respectively. Sample 1 was combined with 25% of the filter area from the two HFO-fueled vessels, namely YK and YF; Sample 2 was combined with 25% of filter area from two diesel-fueled vessels, namely GB1 and TB4; samples 3, 4, and 5 were combined with 50% of the filter areas from four excavators under idling, moving, and working modes, respectively, namely CAT320, CAT330B, CAT307 and PC60."

**_Comment#5:_** Experimental, Filter Collection Details:

What was the flowrate used for $PM_{2.5}$ sampling? Did you have denuders? If not, what potential positive artifacts occurred on your quartz filters when conducting the molecular composition analyses? Don't you expect some absorption of semivolatiles on these filters?

**Response:** The flowrate used for $PM_{2.5}$ sampling in this study was 10 $L \cdot min^{-1}$. The denuders were not used in our study.

Schauer et al., (1999) compared the organic carbon mass emitted from medium-duty diesel trucks between denuder-based sampling technique and traditional filter-based sampling technique. They found that particulate organic carbon emission rate determined by the denuder-based sampling technique was found to be 35% lower than the organic carbon mass collected using a traditional filter-based sampling technique. This was concluded to be a result of a positive vapor-phase sorption artifact that affects the traditional filter sampling technique. It was reported that the quartz filter has a large surface area upon which adsorption of gaseous organics could occur, causing a positive artifact (Cheng et al., 2010). It was reported by Cheng et al., (2010) that in China, positive sampling artifact constituted 10% and 23% of the OC concentration determined by the bare quartz filter during winter and summer, respectively.

However, potential problems that arise from the usage of denuders include incomplete gas-phase removal, particle loss in the denuder tube, and semi-volatile compound off-gassing from particles when their corresponding gas phase components are removed in the denuder. It was reported that 5%-10% of the particles was lost in the denuders (Temime-Roussel et al., 2004). As a result, particle sampling in this study has ceased to use denuders.

Table S3 The most abundant peaks of CHOS compounds emitted from excavators under three operation modes and diesel-fueled vessels.

| | [M-H]$^-$ | m/z | DBE | Relative response (%) | | [M-H]$^-$ | m/z | DBE | Relative response (%) |
|---|---|---|---|---|---|---|---|---|---|
| Excavator under idling | C16H31O5S$^-$ | 335.1898 | 1 | 23.50 | Excavator under moving | C5H3O13S2$^-$ | 334.902 | 4 | 11.95 |
| | C17H33O5S$^-$ | 349.2054 | 1 | 22.42 | | C4H3O11S2$^-$ | 290.9121 | 3 | 3.25 |
| | C15H29O5S$^-$ | 321.1741 | 1 | 22.07 | | C22H37O3S$^-$ | 381.2469 | 4 | 3.02 |
| | C18H35O5S$^-$ | 363.2211 | 1 | 19.07 | | C14H27O5S$^-$ | 307.1585 | 1 | 2.69 |
| | C14H27O5S$^-$ | 307.1585 | 1 | 16.28 | | C15H29O5S$^-$ | 321.1742 | 1 | 2.64 |
| | C17H35O5S$^-$ | 351.2211 | 0 | 16.12 | | C16H31O5S$^-$ | 335.1898 | 1 | 2.48 |
| | C16H29O5S$^-$ | 333.1741 | 2 | 14.63 | | C15H27O5S$^-$ | 319.1585 | 2 | 2.37 |
| | C17H31O5S$^-$ | 347.1898 | 2 | 14.42 | | C18H29O4S$^-$ | 341.1792 | 4 | 2.28 |
| | C18H33O5S$^-$ | 361.2054 | 2 | 14.39 | | C13H25O5S$^-$ | 293.1428 | 1 | 2.07 |
| | C15H27O5S$^-$ | 319.1585 | 2 | 13.89 | | C16H29O5S$^-$ | 333.1741 | 2 | 2.06 |
| Excavator under working | C22H37O3S$^-$ | 381.2469 | 4 | 33.63 | Diesel-fueled vessel | C12H25O5S$^-$ | 281.1428 | 0 | 22.20 |
| | C24H41O3S$^-$ | 409.2782 | 4 | 14.90 | | C13H27O5S$^-$ | 295.1585 | 0 | 18.86 |
| | C5H3O13S2$^-$ | 334.902 | 4 | 11.85 | | C11H23O5S$^-$ | 267.1272 | 0 | 16.00 |
| | C16H29O5S$^-$ | 333.1741 | 2 | 8.43 | | C13H25O5S$^-$ | 293.1428 | 1 | 15.57 |
| | C15H27O5S$^-$ | 319.1585 | 2 | 8.22 | | C15H29O5S$^-$ | 321.1741 | 1 | 15.01 |
| | C16H31O5S$^-$ | 335.1898 | 1 | 7.89 | | C14H27O5S$^-$ | 307.1585 | 1 | 14.75 |
| | C17H31O5S$^-$ | 347.1898 | 2 | 7.70 | | C12H23O5S$^-$ | 279.1272 | 1 | 12.64 |
| | C15H29O5S$^-$ | 321.1741 | 1 | 7.58 | | C11H21O5S$^-$ | 265.1115 | 1 | 11.14 |
| | C17H33O5S$^-$ | 349.2054 | 1 | 7.23 | | C16H31O5S$^-$ | 335.1898 | 1 | 11.03 |
| | C14H27O5S$^-$ | 307.1585 | 1 | 6.77 | | C10H19O5S$^-$ | 251.0959 | 1 | 8.74 |

**Comment#9:** Abstract: The last sentence of the abstract needs to be completely re-worded. The current sentence is poorly worded and not easy to understand.

**Response:** We thank the reviewer for pointing out the lack of clarity. A professional language editing company has thoroughly polished and edited the revised manuscript..

**Comment#10:** I would change "polar organic matters (POM)" to polar organic compounds (POCs).

**Response:** Thanks for your suggestion. The "polar organic matters (POM)" has been changed into "polar organic compounds (POCs) through all of the revised manuscript.

**Comment#11:** Introduction, Page 6, Lines 6-8:

Change this sentence to state:

"This study aimed to chemical characterize polar organic aerosol constituents at the molecular level that are emitted from typical non-road engines by FT-ICR MS to provide new aerosol marker compounds for non-road engines."

**Response:** Thanks. The sentence has been modified as reviewer suggestion (**Page 7 lines 15-17**).

**Comment#12:** Page 7, Line 5: Do you mean to say "plume" instead of "flume"?

**Response:** Thanks. "Flume" has been changed into "plume" (**Page 8 line 19**).

**Comment#13:** Experimental Section, Page 8, Line 14: Delete "continually"

**Response:** Thanks. "Continually" has been deleted (**Page 10 line 22**).

[revised manuscript text omitted]

---

## Referee Report (RR1)

The revised manuscript well addressed most concerns of the two reviewers. However, I still think that it is inappropriate to state that the structures of polar organic compounds were characterized. In fact, only the elemental compositions in organic molecules were inferred. With the same condensed chemical formula, there may be different structures of organics, which may further represent emissions of different sources or secondary formations from different pathways. Thus, I suggest changing the definite expressions on molecular structures to speculative ones throughout the manuscript. Then, the title of the manuscript has been changed to "Molecular Characterization of Polar Organic Aerosol Constituents in Off-Road Engine Emissions Using Fourier Transform Ion Cyclotron Resonance Mass Spectrometry (FT-ICR-MS): Implications for Source Apportionment", according to the suggestion of one reviewer. Correspondingly, the conclusion section has also been expanded, with the new subtitle of "Conclusions and Environmental implications". However, the implications on source apportionment are not clearly presented. Do you find some tracers that are new and specific for some sources? Will the tracers be applicable in source apportionment, given that most findings are about the elemental compositions and DBE values rather than the speciated organic tracers? Will the findings substantially improve the current source apportionment results, like the clear separation of fossil and non-fossil sources by $^{14}$C? For me, the last section reads more like discussions on the results in the present study and previous studies, while the significance and scientific value of the study have not been demonstrated. As a study providing fundamental measurements of source emissions, it is essential to indicate the atmospheric relevance of the findings.

---

## Author Response (AR2)

**Response to the comments**

**Journal:** Atmospheric Chemistry and Physics

**Manuscript ID:** acp-2019-449

**Title:** Molecular Characterization of Polar Organic Aerosol Constitents in Off-Road Engine Emissions Using Fourier Transform Ion Cyclotron Resonance Mass Spectrometry (FT-ICR MS): Implications for Source Apportiopnment

**Author(s):** Min Cui, Cheng Li, Yingjun Chen*, Fan Zhang, Jun Li*, Bin Jiang, Yangzhi Mo, Jia Li, Caiqing Yan, Mei Zheng, Zhiyong Xie, Gan Zhang, and Junyu Zheng*

**Corresponding author:** Yingjun Chen; Jun Li and Junyu Zheng (email: yjchenfd@fudan.edu.cn; junli@gig.ac.cn; zhengjunyu_work@hotmail.com)

Thank you so much for your constructive comments! So far, we have revised the manuscript accordingly. Our point-by-point responses (in black) to each comment are listed below. And the modifications in the revised manuscript with marks are marked in red. Please see the manuscript for details.

*Comment #1:* The revised manuscript well addressed most concerns of the two reviewers. However, I still think that it is inappropriate to state that the structures of polar organic compounds were characterized. In fact, only the elemental compositions in organic molecules were inferred. With the same condensed chemical formula, there may be different structures of organics, which may further represent emissions of different sources or secondary formations from different pathways. **Thus, I suggest changing the definite expressions on molecular structures to speculative ones throughout the manuscript.**

Response: Thank you so much for your kindly suggestion. Due to the limitation of recognizing the isomers of polar organic matters (POCs) for FT-ICR MS, possible structures of POCs were speculated by referencing some empirical values and comparing with structures reported in references (Wozniak et al., 2008; Lin et al.,

2012; Yassine et al., 2014; Riva et al., 2015; Riva et al., 2016). Thus, the definite expressions on molecular structures have been modified to speculative ones throughout the revised manuscript to avoid misleading readers into thinking that it is the accurate structures emitted from non-road engines (**Page 1 line 3, line 15; Page 6 line 14; Page 14 line 22; Page 15 line 4, line 14; Page 18 line 9, line 14**).

**References:**

[1] Wozniak, A. S., Bauer, J. E., Sleighter, R. L., Dickhut, R. M. and Hatcher, P. G.: Technical Note: Molecular characterization of aerosol-derived water-soluble organic carbon using ultrahigh resolution electrospray ionization Fourier transform ion cyclotron resonance mass spectrometry, Atmospheric Chemistry and Physics, 8 (17): 5099-5111, 2008.

[2] Lin, P., Rincon, A. G., Kalberer, M. and Yu, J. Z.: Elemental Composition of HULIS in the Pearl River Delta Region, China: Results Inferred from Positive and Negative Electrospray High Resolution Mass Spectrometric Data, Environmental Science & Technology, 46 (14): 7454-7462, 2012.

[3] Yassine, M. M., Harir, M., Dabek-Zlotorzynska, E. and Schmitt-Kopplin, P.: Structural characterization of organic aerosol using Fourier transform ion cyclotron resonance mass spectrometry: Aromaticity equivalent approach, Rapid Communications in Mass Spectrometry, 28 (22): 2445-2454, 2014.

[4] Riva, M., Tomaz, S., Cui, T., Lin, Y. H., Perraudin, E., Gold, A., Stone, E. A., Villenave, E., Surratt, J. D.: Evidence for an unrecognized secondary anthropogenic source of organosulfates and sulfonates: gas-phase oxidation of polycyclic aromatic hydrocarbons in the presence of sulfate aerosol, Environ Sci Technol, 49(11): 6654-6664, 2015.

[5] Riva, M., Da Silva Barbosa, T., Lin, Y. H., Stone, E. A., Gold, A., Surratt, J. D.: Characterization of organosulfates in secondary organic aerosol derived from the photooxidation of long  chain alkanes. Atmospheric Chemistry and Physics:1-39, 2016.

*Comment #2:* Then, the title of the manuscript has been changed to "Molecular Characterization of Polar Organic Aerosol Constituents in Off-Road Engine Emissions Using Fourier Transform Ion Cyclotron Resonance Mass Spectrometry (FT-ICR-MS): Implications for Source Apportionment", according to the suggestion of one reviewer. Correspondingly, the conclusion section has also been expanded, with the new subtitle of "Conclusions and Environmental implications". **However, the implications on source apportionment are not clearly presented.** Do you find some tracers that are new and specific for some sources? Will the tracers be applicable in source apportionment, given that most findings are about the elemental compositions and DBE values rather than the speciated organic tracers? Will the findings substantially improve the current source apportionment results, like the clear separation of fossil and non-fossil sources by 14C? For me, the last section reads more like discussions on the results in the present study and previous studies, while the significance and scientific value of the study have not been demonstrated. **As a study providing fundamental measurements of source emissions, it is essential to indicate the atmospheric relevance of the findings.**

**Response:** We are highly appreciated of the constructive suggestion. Some atmospheric implications were discussed as follows and added in the revised manuscript.

On the one hand, some characteristic compounds (e.g. condensed aromatic ring with high O numbers and nitronaphthol with one or more methyl groups in HFO-fueled vessels; dinitrophenol in non-road diesel vehicle emissions) were speculated from non-road engine emissions through comparing the probable chemical structures of POCs from different sources in part of Discussion and environmental implications in the revised manuscript. These compounds could provide implication for source identification research in the future, especially using FT-ICR MS (**Page 24 lines 7-22; Page 25 lines 1-22; Page 26 lines 1-6**).

On the other hand, the most important finding in this research was that S-containing compounds with condensed aromatic rings potentially was unique tracers for heavy-oil fueled (HFO) vessels emission. HFO-fueled vessels emission is one of the

most important sources of PM, especially in port area. The contribution of these vessels emission to atmospheric particulate matters was hardly to evaluate accurately due to lacking of unique tracers to distinguish emissions from HFO-fueled vessels and other sources. Thus, the findings in this research could significantly fill the cognitive gaps. Furthermore, some quantitative methods based on FT-ICR MS have been developed to semi-quantify polar compounds such as sulfur-containing compounds (Jiang et al., 2019; Muller et al., 2012; Zhou et al., 2016). In the future, evaluating the contribution of HFO vessels emission to atmospheric particulate might come true by quantifying the concentrations of these S-containing compounds with condensed aromatic rings emitted from different sources and in the atmosphere.

Finally, these atmospheric implications were emphasized in the part of Discussion and environmental implications in the revised manuscript (**Page 26 lines 7-14**).

[revised manuscript text omitted]

---

## Author Response (AR3)

**Response to the editor's comments**

**Journal:** Atmospheric Chemistry and Physics

**Manuscript ID:** acp-2019-449

**Title:** Molecular Characterization of Polar Organic Aerosol Constituents in Off-Road Engine Emissions Using Fourier Transform Ion Cyclotron Resonance Mass Spectrometry (FT-ICR MS)

**Author(s):** Min Cui, Cheng Li, Yingjun Chen*, Fan Zhang, Jun Li*, Bin Jiang, Yangzhi Mo, Jia Li, Caiqing Yan, Mei Zheng, Zhiyong Xie, Gan Zhang, and Junyu Zheng*

**Corresponding author:** Yingjun Chen; Jun Li and Junyu Zheng (email: yjchenfd@fudan.edu.cn; junli@gig.ac.cn; zhengjunyu_work@hotmail.com)

Thank you so much for your constructive suggestions! So far, we have revised the manuscript accordingly. Our point-by-point responses (in black) to each comment are listed below. And the modifications in the revised manuscript with marks are marked in red. Please see the revised manuscript for details.

*Comment #1:* I thank the authors for tempering their language when referring to proposed molecular structures, however they authors should take care when using words like 'probably' because this word implies some kind of statistical likelihood. In this instance, the molecules are based on a combination of previous literature and the authors' judgement, so the text should more clearly reflect this. In particular, the opening sentence of the abstract would be more appropriate as: "The molecular compositions of polar organic compounds (POCs) in particles emitted from various vessels and excavators were characterized using Fourier Transform Ion Cyclotron Resonance Mass Spectrometry (FT-ICR MS) and possible molecular structures proposed."

**Response:** Thank you so much for your kindly suggestion. After carefully reviewing the meaning of "probably", we realized that this word indicates something that is of considerable certainty on the basis of available evidence. We agree with editor's

suggestion that the words like "possible" are more appropriate, hence, the opening sentence of the abstract has been modified (**Page 2 lines 3-6; Page 17 line 16; Page 18 lines 3-4. Page 18 lines 10-12; Page 21 line 12**). Furthermore, words like "conclude", "propose" or "refer" were added in the revised manuscript to reflect the conclusions obtained based on a combination of previous literature and our own findings (**Page 2 lines 20-21; Page 14 line 19; Page 20 line 17; Page 21 line 10; Page 21 line 16**).

**Revision was made as follows:**

(1) "*The molecular compositions and probable structures of polar organic compounds (POCs) in particles emitted from various vessels and excavators were characterized using Fourier Transform Ion Cyclotron Resonance Mass Spectrometry (FT-ICR MS) and **possible** molecular structures of POCs were proposed.*"

(2) "*The composition and structure of the S-containing compounds were directly influenced by fuel oil characteristics (sulfur content and aromatic ring), with more condensed aromatic rings in the S-containing compounds **proposed** in HFO-fueled vessel emissions. More abundant aliphatic chains were **inferred** in diesel equipment emissions.*"

(3) "*Upon comparing the ratios of H/C and O/C for CHO compounds for different off-road engines under three operational modes and using different fuel oils, we **concluded** that the CHO group for vessels using HFO had the highest degree of oxidation and unsaturation.*"

(4) "*For further discussion of **possible** chemical structures, the CHON group was divided into 23 subgroups, including OxN1 (1 $\leqslant$ x $\leqslant$ 10) and OyN2 (2 $\leqslant$ y $\leqslant$ 14).*"

(5) "*When the value of Xn exceeds 2.5, aromatic structures are **most likely to** be present within the compounds*"

(6) "*It **most likely** indicated the presence of a benzene core structure in the compounds. Thus, it **could be** dinitrophenol, and methyl dinitrophenol compounds.*"

(7) "*Upon comparing the sulfur-containing products observed in this study and the study by Riva et al., it was interesting to **conclude** that some PAH-derived OS products generated in the lab also have a significant response in field*"

*measurements.*"

(8) *"Through a comparison of the average DBE value and fraction of compounds with $Xn \geqslant 2.5$ between excavators and vessels, it was **proposed** that different structures were present in S-containing compounds."*

(9) *"The most abundant S-containing compounds emitted by off-road diesel engines were **possibly** aliphatic with long chains and sulfate fraction."*

(10) *"It was interesting to **infer** that the most abundant peaks of CHOS compounds observed in this study were also identified through the laboratory simulation study."*

**Comment #2:** The final paragraph of the discussions and conclusions section is vague and the quality of English is so low as to be unreadable. This must be rewritten.

**Response:** Sorry to make the final paragraph of the discussions vague and unreadable. The new implication was rewritten, which could be seen through the comments #3. And the English grammar of the atmospheric implication has been polished by a professional language editing company.

**Comment #3:** The title states that the paper will cover "implications for source apportionment" yet this is still not adequately discussed. To whit, the word 'apportionment' appears nowhere in the body of the text. In order for this paper to be considered in-scope for ACP, this must be discussed properly, giving examples of the type of source apportionment work that this work would influence. This could be added to the revised discussion section but I would expect more than one paragraph would be required to do this.

**Response:** Thank you for your constructive suggestions aimed at increasing the impact of our research. Because of the suggestions given by the editor and reviewer, the detailed implications of this research have been added in the revised manuscript (**Page 26 lines 18-22; Page 27 lines 1-22; Page 28 lines 1-12**). However, to prevent confusion among readers, we decided to modify the title to "Molecular Characterization of Polar Organic Aerosol Constituents in Off-Road Engine Emissions Using Fourier Transform Ion Cyclotron Resonance Mass Spectrometry (FT-ICR MS)."

**Revision was made as follows:**

"*In recent years, air pollution from ship emissions, especially in coastal areas, has drawn increasing worldwide attention. In previous studies, vanadium (V) and nickel (Ni) were widely used as specific tracers for heavy fuel oil (HFO) fueled vessel emissions (Liu et al., 2017). It was reported that the V content in HFO gradually decreased from 39.5 ppm in 2013 to 12.7 ppm in 2018 in China in compliance with the requirements for all vessels within domestic emission control areas (DECA) (Zhang et al., 2019). In addition, as V and Ni could also originate from industrial emissions, the uncertainties in estimating ship emission contributions to atmospheric $PM_{2.5}$ could increase if only these heavy metals are used as tracers (Zhang et al., 2014). Meanwhile, the relative intensity of OC components from HFO-fueled vessel emissions increased from 10.9% to 21.7% after the implementation of DECA policy (Zhang et al., 2019), while the sulfur content of some heavy oils remained higher than those regulated by oil standards of ships (Zhang et al., 2018). Therefore, probing the molecular-level characteristics of S-rich organic matter from HFO-fueled vessel emissions will be effective and useful.*

*In this study, we found that the molecular compositions of POCs emitted by HFO-fueled vessels were different from those of other source emissions (e.g. off-road diesel engines, biomass and coal burning). The results of this study demonstrated that soot-materials or oxidized polycyclic aromatic hydrocarbons and S-containing species especially those with high aromaticity and O/S ratios most likely organosulfates or sulfonates constitute significant fractions of POCs in HFO-fueled vessel smoke, and were significantly different from those in other primary source emissions. Therefore, it provides an opportunity to act as potential molecular markers to distinguish HFO-fueled vessel emissions in the future.*

*On the other hand, it should be noted that these organosulfates and sulfonates have been previously reported to be mainly formed from secondary photochemical reactions via the oxidation products of VOCs and acidified sulfate seed particles or sulfuric acid in atmosphere (Riva et al., 2015; Tao et al., 2014). However, the high abundance of S-containing species found in HFO-fueled vessel smoke indicates that not only secondary*

*organic aerosols but also primary HFO-fueled vessel smoke could be an important source of organosulfates (**Figure S5**). Therefore, ignoring the contributions of HFO-fueled vessel emissions to organosulfates might lead to the overestimation of the contribution of secondary organic aerosols in the atmosphere.*

*Although some useful characterizations of POCs from off-road engine combustion emissions were proposed, some issues still need to be resolved in the future. This include, (1) determination of the molecular structure of the distinctive compounds mentioned in this study should be further explored; and (2) potentially different molecular structures of organosulfates from HFO-fueled vessel emissions and SOA should also be distinguished.*"

**Comment #4:** In addition, I spotted the following typos in the process of checking this version, although there may be others, so I would recommend the authors perform an additional proofread:

Page 18, line 9: Change 'probable' to 'probably'.

Page 25, line 2: Change 'om' to 'on'

**Response:** Thank you. The typos reminded by editor have been modified. And the proofread was conducted throughout the revised manuscript (**Page 4 line 21; Page 6 line 2; Page 12 line 5; Page 15 line 1; Page 18 line 9; Page18 line 15; Page 19 line 1; Page 23 line 5; Page 34 lines 11-13**).

设置了格式: 下标

设置了格式: 下标